# Binomial Gradient-Based Meta-Learning for Enhanced Meta-Gradient Estimation

**Yilang Zhang**[*1], **Abraham Jaeger Mountain**[*1], **Bingcong Li**[2] **& Georgios B. Giannakis**[1]

[1]Department of Electrical and Computer Engineering
University of Minnesota
Minneapolis, MN 55455, USA
`{zhan7453,jaege252,georgios}@umn.edu`

[2]Department of Computer Science
ETH Zürich
8092 Zürich, Switzerland
`{bingcong.li}@inf.ethz.ch`

## Abstract

Meta-learning offers a principled framework leveraging *task-invariant* priors from related tasks, with which *task-specific* models can be fine-tuned on downstream tasks, even with limited data records. Gradient-based meta-learning (GBML) relies on gradient descent (GD) to adapt the prior to a new task. Albeit effective, these methods incur high computational overhead that scales linearly with the number of GD steps. To enhance efficiency and scalability, existing methods approximate the gradient of prior parameters (meta-gradient) via truncated backpropagation, yet suffer large approximation errors. Targeting accurate approximation, this work puts forth binomial GBML (BinomGBML), which relies on a truncated binomial expansion for meta-gradient estimation. This novel expansion endows more information in the meta-gradient estimation via efficient parallel computation. As a running paradigm applied to model-agnostic meta-learning (MAML), the resultant BinomMAML provably enjoys error bounds that not only improve upon existing approaches, but also decay super-exponentially under mild conditions. Numerical tests corroborate the theoretical analysis and showcase boosted performance with slightly increased computational overhead.

## 1 Introduction

Deep learning (DL) has well-documented success in recent years across various domains (Vaswani, 2017; He et al., 2016; Brown et al., 2020). With its merits granted, the effectiveness of DL relies markedly on high-dimensional models that require massive datasets for training. In applications where data are scarce, acquiring massive training datasets can be prohibitively expensive. Examples of such "data-limited" regimes include medical imaging (Varoquaux & Cheplygina, 2022), social sciences (Grimmer et al., 2022), and robot manipulation (Kroemer et al., 2021).

As opposed to data-hungry DL, a hallmark of human intelligence is the ability to exploit past knowledge to swiftly acquire new skills through limited experiences and trials. *Meta-learning* aims to impart humans' knowledge-generalizing ability into DL by identifying some *task-invariant* prior knowledge, which can be subsequently transferred to aid the learning of new tasks, even when training data are limited. Akin to human learning from past experiences, meta-learning accumulates the sought prior knowledge from a set of related tasks. Model-based approaches, which attempt to encode prior task knowledge in some additional meta-NN, have seen some success, but are prone to poor robustness and require hand-tooled design.

In contrast, gradient-based meta-learning (GBML) uses an iterative optimizer, such as gradient descent (GD), to train task-specific models, with prior information embedded in the learnable hyperparameters of the optimizer. By learning efficient optimization hyperparameters, the model can rapidly adapt to

---

new tasks in a few optimization iterations (Andrychowicz et al., 2016). A representative example is model-agnostic meta-learning (MAML) (Finn et al., 2017), which encodes the prior using an initialization shared across tasks and relies on GD to adapt the task-specific models. By starting from an informative initial point, the model can easily converge to some local minimum even with limited training data. Building upon MAML, many GBML variants have been developed to further improve performance (Yoon et al., 2018; Khodak et al., 2019; Nichol, 2018; Raghu et al., 2019).

While GBML has gained popularity in various application domains (Zhu et al., 2020; Zang et al., 2020; Chen et al., 2021) the process of learning the prior can incur high computational complexity. In particular, the complexity for calculating the gradient of prior (a.k.a. meta-gradient) is linear with the number of adaptation steps, rendering these algorithms barely scalable. To lower the computational overhead of GBML, meta-gradient estimators have been investigated as an alternative to vanilla full backpropagation. For instance, first-order approaches (Finn et al., 2017; Nichol, 2018) do not make explicit use of higher-order information. However, this "over-simplification" can lead to large estimation errors, thereby decelerating meta-learning convergence and resulting in degraded performance. For a desirable trade-off between accuracy and complexity, truncated backpropagation (Shaban et al., 2019) is advocated, in the more general context of gradient-based bilevel optimization, to retain a specific portion of second-order information. Another line of work (Rajeswaran et al., 2019; Ravi & Beatson, 2019; Zhang et al., 2023) utilizes implicit differentiation to replace full backpropagation, but the implicit function theorem (Krantz & Parks, 2002) involved relies markedly on the stationarity of the solution found by the iterative optimizer.

While first-order and truncated backpropagation lessen the computational load of GBML, we will demonstrate with analysis and empirical observations that they can lead to large approximation errors. With this motivation, we revisit the GBML meta-gradient formulation and develop a novel meta-gradient estimate, binomial gradient-based meta-learning (BinomGBML), which earns its name from the keystone binomial theorem. Our contribution is three-fold:

- Embracing efficient parallelization, BinomGBML improves upon state-of-the-art meta-gradient estimators by retaining more information to markedly reduce the estimation error.

- Theoretical analyses under three different assumptions reveal BinomMAML's reduced estimation error, which decays super-exponentially. Extensive numerical tests on synthetic and real datasets further validate these claims.

- As a side benefit, when combined with MAML (Finn et al., 2017), BinomMAML addresses the memory scalability of MAML through dynamic management of computational graphs.

**Notation.** Bold lowercase (uppercase) letters denote vectors (matrices); $\| \cdot \|$ and $\langle \cdot, \cdot \rangle$ stand for $\ell_2$-norm and inner product; and $[\cdot]_i$ for the $i$-th entry (column) of a vector (matrix).

## 2 PRELIMINARIES

### 2.1 PROBLEM STATEMENT

Meta-learning first acquires a task-invariant prior parameterized by $\boldsymbol{\theta} \in \mathbb{R}^D$ that is common to a set of related tasks. Let $t = 1, \ldots, T$ index these tasks, and $\mathcal{D}_t := \{(\mathbf{x}_t^n, y_t^n)\}_{n=1}^{N_t}$ denote the per-task datasets comprising $N_t$ (limited) input-label pairs. For compactness, let matrix $\mathbf{X}_t \in \mathbb{R}^{d \times N_t}$ be formed with columns $\mathbf{x}_t^n$, and $\mathbf{y}_t \in \mathbb{R}^{N_t}$ be the corresponding label vector. Each $\mathcal{D}_t$ is partitioned into a training subset $\mathcal{D}_t^{\mathrm{trn}} \subset \mathcal{D}_t$ and a validation subset $\mathcal{D}_t^{\mathrm{val}} := \mathcal{D}_t \setminus \mathcal{D}_t^{\mathrm{trn}}$. Instead, the meta-training process first extracts a task-invariant prior from these tasks by leveraging their associated data. Then, to measure the quality of the meta-trained prior, an unseen meta-testing task indexed by $\star$ is provided with limited training data $\mathcal{D}_\star^{\mathrm{trn}}$, and some test inputs $\{\mathbf{x}_\star^n\}$. As $N_\star$ can be markedly smaller than required for DL, directly training a model over $\mathcal{D}_\star^{\mathrm{trn}}$ could easily lead to overfitting. The meta-testing process transfers the acquired prior to the new task, at which point a DL model can be fine-tuned over the small $\mathcal{D}_\star^{\mathrm{trn}}$ to predict the corresponding test labels $\{y_\star^n\}$.

The meta-training process amounts to a nested empirical risk minimization problem. The inner level (task level) trains the per-task model parameter $\boldsymbol{\phi}_t \in \mathbb{R}^d$ over $\mathcal{D}_t^{\mathrm{trn}}$ using the task-invariant prior provided by the outer level. The outer level (meta level) assesses the trained model on $\mathcal{D}_t^{\mathrm{val}}$, and adjusts the prior accordingly. Recall $\boldsymbol{\theta} \in \mathbb{R}^D$ is the shared prior parameter (a.k.a. meta-parameter).

This nested structure is naturally expressed as a bilevel objective

$$\min_{\boldsymbol{\theta}} \ \sum_{t=1}^{T} \ell_t^{\mathrm{val}}(\boldsymbol{\phi}_t^*(\boldsymbol{\theta})) \tag{1a}$$

$$\text{s.t. } \boldsymbol{\phi}_t^*(\boldsymbol{\theta}) = \arg\min_{\boldsymbol{\phi}_t} \ell_t^{\mathrm{trn}}(\boldsymbol{\phi}_t) + \mathcal{R}(\boldsymbol{\phi}_t; \boldsymbol{\theta}) \tag{1b}$$

where $\ell_t^{\mathrm{trn}}(\boldsymbol{\phi}_t) := \ell^{\mathrm{trn}}(\boldsymbol{\phi}_t; \mathcal{D}_t^{\mathrm{trn}}) := -\log p(\mathbf{y}_t^{\mathrm{trn}}|\boldsymbol{\phi}_t; \mathbf{X}_t^{\mathrm{trn}})$ is the training loss (negative log-likelihood) evaluating the fit of the task-specific model to $\mathcal{D}_t^{\mathrm{trn}}$, and regularizer $\mathcal{R}(\boldsymbol{\phi}_t; \boldsymbol{\theta}) := -\log p(\boldsymbol{\phi}_t; \boldsymbol{\theta})$ represents the negative log-prior probability density function. Using Bayes' rule, one can verify that $\boldsymbol{\phi}_t^*(\boldsymbol{\theta}) = \arg\min_{\boldsymbol{\phi}_t} -\log p(\boldsymbol{\phi}_t|\mathcal{D}_t^{\mathrm{trn}}; \boldsymbol{\theta})$ is the maximum a posteriori estimator.

While formulation (1) is appealing, the global optimum $\boldsymbol{\phi}_t^*$ in (1b) is generally intractable for highly nonlinear NNs. A feasible alternative is to rely on a manageable solver to approximate $\boldsymbol{\phi}_t^*$.

Gradient-based meta-learning (GBML) poses the solver as a few cascaded optimization steps. The meta-parameter $\boldsymbol{\theta}$ acts as the optimizer hyperparameters shared among all tasks. The first work along this line is termed model-agnostic meta-learning (MAML) (Finn et al., 2017). It relies on a $K$-step GD optimizer as the solver, with a task-invariant initialization being the meta-parameter (thus $D = d$), which renders the alternative objective

$$\min_{\boldsymbol{\theta}} \ \frac{1}{T} \sum_{t=1}^{T} \mathcal{L}_t(\boldsymbol{\theta}) := \frac{1}{T} \sum_{t=1}^{T} \ell_t^{\mathrm{val}}(\boldsymbol{\phi}_t^K(\boldsymbol{\theta})) \tag{2a}$$

$$\text{s.t. } \boldsymbol{\phi}_t^0(\boldsymbol{\theta}) = \boldsymbol{\theta}, \ \ \boldsymbol{\phi}_t^{k+1}(\boldsymbol{\theta}) = \boldsymbol{\phi}_t^k(\boldsymbol{\theta}) - \alpha \nabla \ell_t^{\mathrm{trn}}(\boldsymbol{\phi}_t^k(\boldsymbol{\theta})), \ k = 0, \ldots, K-1 \tag{2b}$$

where $\alpha > 0$ is a constant learning rate. Although (2) contains no explicit $\mathcal{R}$, it is pointed out that under second-order approximation (Grant et al., 2018), (2b) satisfies

$$\boldsymbol{\phi}_t^K(\boldsymbol{\theta}) \approx \boldsymbol{\phi}_t^*(\boldsymbol{\theta}) = \arg\min_{\boldsymbol{\phi}_t} \ell_t^{\mathrm{trn}}(\boldsymbol{\phi}_t) + \frac{1}{2}\|\boldsymbol{\phi}_t - \boldsymbol{\theta}\|_{\boldsymbol{\Lambda}_t}^2$$

where $\boldsymbol{\Lambda}_t$ is determined by $\alpha$, $K$, and $\nabla^2 \ell_t^{\mathrm{trn}}(\boldsymbol{\theta})$. In other words, MAML's optimization strategy is equivalent to an implicit Gaussian prior $p(\boldsymbol{\phi}_t; \boldsymbol{\theta}) = \mathcal{N}(\boldsymbol{\phi}_t; \boldsymbol{\theta}, \boldsymbol{\Lambda}_t)$, whose mean is the task-invariant initialization.

## 2.2 COMPUTATIONAL CHALLENGES IN META-LEARNING

This section examines the time and space complexities of GBML. For illustration, the discussion will particularly focus on MAML, while similar analysis can be readily applied to other GBML methods including (Li et al., 2017; Lee & Choi, 2018; Rusu et al., 2018; Park & Oliva, 2019).

Optimizing $\boldsymbol{\theta}$ in (2a) requires computing the gradient

$$\nabla \mathcal{L}_t(\boldsymbol{\theta}) = \nabla_{\boldsymbol{\theta}} \ell_t^{\mathrm{val}}(\boldsymbol{\phi}_t^K(\boldsymbol{\theta})) \overset{(a)}{=} \prod_{k=0}^{K-1} \nabla \boldsymbol{\phi}_t^k(\boldsymbol{\phi}_t^{k-1}) \nabla \ell_t^{\mathrm{val}}(\boldsymbol{\phi}_t^K) \overset{(b)}{=} \prod_{k=0}^{K-1} \left[ \mathbf{I}_d - \alpha \nabla^2 \ell_t^{\mathrm{trn}}(\boldsymbol{\phi}_t^k) \right] \nabla \ell_t^{\mathrm{val}}(\boldsymbol{\phi}_t^K) \tag{3}$$

where $(a)$ utilizes the chain rule, and $(b)$ is from (2b). For compactness, let $\mathbf{H}_t^k := \nabla^2 \ell_t^{\mathrm{trn}}(\boldsymbol{\phi}_t^k)$ denote the Hessian, and $\mathbf{g}_t^K := \nabla \ell_t^{\mathrm{val}}(\boldsymbol{\phi}_t^K)$ the validation gradient. It follows from (3) that

$$\nabla \mathcal{L}_t(\boldsymbol{\theta}) = \prod_{k=0}^{K-1} \left[ \mathbf{I}_d - \alpha \mathbf{H}_t^k \right] \mathbf{g}_t^K. \tag{4}$$

For any vector $\mathbf{g} \in \mathbb{R}^d$, one can compute $[\mathbf{I}_d - \alpha \mathbf{H}_t^k]\mathbf{g}$ by first expanding $[\mathbf{I}_d - \alpha \mathbf{H}_t^k]\mathbf{g} = \mathbf{g} - \alpha \mathbf{H}_t^k \mathbf{g}$ and then using the Hessian-vector product (HVP) to efficiently obtain $\mathbf{H}_t^k \mathbf{g} = \nabla_{\boldsymbol{\phi}_t^k}\langle \nabla \ell_t^{\mathrm{trn}}(\boldsymbol{\phi}_t^k), \mathbf{g} \rangle$. As a result, $\mathcal{O}(Kd)$ time and space are required to compute (4) through backpropagation.

Note that acquiring $\boldsymbol{\phi}_t^K$ via (2b) also incurs $\mathcal{O}(Kd)$ time complexity, but the constant hidden inside $\mathcal{O}$ is much smaller than (3). Thus, the overall time complexity is dominated by the backpropagation process (Griewank, 1993). Given that GD converges sub-linearly when the gradient is Lipschitz, it

requires $K = \mathcal{O}(1/\epsilon)$ iterations for $\phi_t^K$ to reach an $\epsilon$-stationary point (Bertsekas, 2016). This implies that a sufficiently large $K$ is required to accurately solve (2), raising concerns about the scalability of MAML as its time and space complexities both grow linearly with $K$.

To address this issue, first-order (FO)MAML (Finn et al., 2017) was proposed alongside 'vanilla MAML,' providing a simpler but less accurate approximation to $\nabla \mathcal{L}_t(\boldsymbol{\theta})$. FOMAML discards all second-order information by simply setting $\mathbf{H}_t^k = \mathbf{0}_{d \times d}, \ \forall \, k$, thus resulting in the $\mathcal{O}(d)$ time and space estimate

$$\hat{\nabla}^{\mathrm{FO}} \mathcal{L}_t(\boldsymbol{\theta}) = \mathbf{g}_t^K.$$

This "brutal" simplification heavily prioritizes reduction of computations at the expense of estimation accuracy, which can degrade model quality and slow meta-training convergence (Sow et al., 2022). A different first-order approach, Reptile (Nichol, 2018), proposes the meta-update rule $\boldsymbol{\theta} \leftarrow \boldsymbol{\theta} + \epsilon \frac{1}{T} \sum_{t=1}^{T} (\phi_t^K - \boldsymbol{\theta})$, with $\epsilon \in (0, 1]$, rather than performing GD on $\mathcal{L}(\boldsymbol{\theta})$. This method implicitly retains higher-order information, but suffers from sensitivity to training setup in practice and, under favorable conditions, results in comparable performance to FOMAML.

To address the limitations of first-order methods, Truncated MAML (TruncMAML) (Shaban et al., 2019) suggests retaining a portion of second-order information by truncating the backpropagation (4). Upon setting $\mathbf{H}_t^k = \mathbf{0}, \ k = 0, 1, \ldots, K - L - 1$, where $L \in [0, K]$ is a user-defined truncation constant, the gradient estimate boils down to

$$\hat{\nabla}^{\mathrm{Tr}} \mathcal{L}_t(\boldsymbol{\theta}) = \prod_{k=K-L}^{K-1} \left[ \mathbf{I}_d - \alpha \mathbf{H}_t^k \right] \mathbf{g}_t^K. \tag{5}$$

Since (5) involves $L \leq K$ HVP computations, the time and space complexities of TruncMAML both decrease to $\mathcal{O}(Ld)$. It is seen that TruncMAML reduces to FOMAML when $L = 0$, and recovers vanilla MAML when $L = K$. It is empirically observed, however, that the gradient error of TruncMAML decreases slowly when $L$ is small; cf. Figure 3b. More rigorous performance analyses will be provided in Section 3.2, with error bounds illustrated in Figure 2. It will turn out that large $L$ sufficiently close to $K$ will be needed to ensure an accurate meta-gradient estimate.

Another approach for meta-gradient estimation, termed implicit MAML (iMAML), is to rely on the implicit function theorem (Krantz & Parks, 2002), which yields a closed-form solution to $\nabla \mathcal{L}_t(\phi_t^*)$. For instance, it has been shown in (Rajeswaran et al., 2019) that, with an explicit regularizer $\mathcal{R}(\phi_t; \boldsymbol{\theta}) = \frac{\lambda}{2} \|\phi_t - \boldsymbol{\theta}\|^2$ induced by an isotropic Gaussian prior, the per-task validation gradient is $\nabla_{\boldsymbol{\theta}} \ell_t^{\mathrm{val}}(\phi_t^*(\boldsymbol{\theta})) = \left[ \mathbf{I}_d + \frac{1}{\lambda} \nabla^2 \ell_t^{\mathrm{trn}}(\phi_t^*(\boldsymbol{\theta})) \right]^{-1} \nabla \ell_t^{\mathrm{val}}(\phi_t^*(\boldsymbol{\theta}))$. Similarly, as the global optimum $\phi_t^*$ is intractable, it must be approximated with a nearly optimal $\phi_t^K$ in the calculation. Moreover, the inverse Hessian-vector product requires another approximate solver, such as conjugate gradient iterations (Rajeswaran et al., 2019). As memory does not grow with the number of solver iterations (denoted $L$), iMAML has time complexity $\mathcal{O}(Ld)$ and memory complexity $\mathcal{O}(d)$. However, the iterative solver introduces additional errors in the meta-gradient estimation, making the algorithms numerically less stable than FOMAML and TruncMAML (Rajeswaran et al., 2019; Zhang et al., 2023). Consequently, reverse-mode explicit automatic differentiation (i.e., backpropagation) has become more prevalent in practice, and this work will primarily focus on it.

# 3 META-GRADIENT VIA TRUNCATED BINOMIAL EXPANSION

While they reduce computational complexity compared to vanilla MAML, FOMAML and Trunc-MAML can suffer from high estimation errors that impair the meta-learning convergence and performance. Our motivation stems from the observation that both (3) and (5) entail a sequential chain of HVP computations that cannot be parallelized. To enhance the accuracy of meta-gradient estimation, our key idea is to incorporate more information by performing parallel computations concurrent with each HVP.

To this end, we will first develop a meta-gradient estimator by truncating the binomial expansion of (4). This leads to an approach that we naturally term binomial gradient-based meta-learning (BinomGBML). In the ensuing Section 3.1, the truncated expansion will be reformulated as a cascade of $L$ vector operators, each corresponding to several parallel HVP computations. Then, Section 3.2 will provably show that BinomGBML not only leads to low estimation error bounds under various

assumptions, but also diminishes at a *super-exponential* rate as $L$ increases, as opposed to the slow decay of TruncGBML. Consequently, a small $L$ suffices for accurate gradient estimation.

Again, the discussion will be exclusive to MAML for simplicity, while our approach can be broadened to other GBML algorithms, along the lines adopted by FOMAML and TruncMAML extensions. All proofs of this section are deferred to the Appendix.

### 3.1 META-GRADIENT ESTIMATION VIA BINOMIAL EXPANSION

Our approach is based on the binomial expansion of (4). Recall the binomial theorem (Beyer, 1984)

$$(1 + z)^K = \sum_{l=0}^{K} \binom{K}{l} z^l, \ \forall z \in \mathbb{R}. \tag{6}$$

This expansion can be generalized to multivariate product $\prod_{i=1}^{n}(1 + z_i)$ and even the matrix product in (4) to yield

$$\prod_{k=0}^{K-1} \left[ \mathbf{I}_d - \alpha \mathbf{H}_t^k \right] = \mathbf{I}_d + \sum_{l=1}^{K} \sum_{0 \le k_{1:l}\uparrow < K} \prod_{i=1}^{l} (-\alpha \mathbf{H}_t^{k_i}). \tag{7}$$

where we define $\{0 \le k_{1:l} \uparrow < K\} := \{0 \le k_1 < k_2 < \ldots < k_l < K\}$ as a combinatorial set containing $\binom{K}{l}$ terms. In (7), $\mathbf{I}_d$ corresponds to the $l = 0$ term $\binom{K}{0} z^0 = 1$ in (6); the summation over $0 \le k_{1:l} \uparrow < K$ and product $\prod_{i=1}^{l}(-\alpha \mathbf{H}_t^{k_i})$ respectively align with the binomial factor $\binom{K}{l}$ and the $l$-th power $z^l$ in (6). Intuitively, if the learning rate $\alpha$ is sufficiently small, the product $\prod_{i=1}^{l}(-\alpha \mathbf{H}_t^{k_i}) = \mathcal{O}(\alpha^l)$ can decrease exponentially with $l$. This motivates our meta-gradient estimator based on the truncated binomial expansion as

$$\hat{\nabla}^{\mathrm{Bi}} \mathcal{L}_t(\boldsymbol{\theta}) = \left[ \mathbf{I}_d + \sum_{l=1}^{L} \sum_{0 \le k_{1:l}\uparrow < K} \prod_{i=1}^{l} (-\alpha \mathbf{H}_t^{k_i}) \right] \mathbf{g}_t^K. \tag{8}$$

As the ignored higher-order terms can be as small as $\mathcal{O}(\alpha^{L+1})$, the gradient error is expected to diminish exponentially as truncation $L$ increases. This expansion is also amenable to parallelization. With $L = 1$ for instance, it follows from (8) that $\hat{\nabla}^{\mathrm{Bi}} \mathcal{L}_t(\boldsymbol{\theta}) = \mathbf{g}_t^K - \alpha \sum_{k=0}^{K-1} \mathbf{H}_t^k \mathbf{g}_t^K$, where the $K$ HVPs in the summation can be efficiently performed in parallel thanks to their computational independence. However, the double sum in (8) involves a total of $\sum_{l=1}^{L} \binom{K}{l}$ terms. Directly computing each of them using parallel HVPs can be prohibitive. To improve efficiency, we advocate reducing redundancy in the summation by merging common terms of different matrix products; see Appendix A.1 for an illustrative example. Using this idea, the next Proposition shows that the truncated binomial expansion can be rewritten as a cascade of matrix operators.

**Proposition 3.1** (matrix operator $\mathbb{B}_t^i$). *Let $\mathbf{P}_t^0 := \mathbf{I}_d$, $\mathbf{P}_t^i := \prod_{k=K-i}^{K-1}(\mathbf{I}_d - \alpha \mathbf{H}_t^k)$, $i = 1, \ldots, K$, and dummy index $k_0 = -1$. Define operator $\mathbb{B}_t^i : \mathbb{R}^{d \times d} \mapsto \mathbb{R}^{d \times d}$ such that for any matrix $\mathbf{M} \in \mathbb{R}^{d \times d}$, $\mathbb{B}_t^i \mathbf{M} := \mathbf{P}_t^i - \alpha \sum_{k_{L-i}=k_{L-1-i}+1}^{K-1-i} \mathbf{H}_t^{k_{L-i}} \mathbf{M}$, $i = 0, \ldots, L-1$. For $0 \le L \le K$, it holds that*

$$\mathbf{I}_d + \sum_{l=1}^{L} \sum_{0 \le k_{1:l}\uparrow < K} \prod_{i=1}^{l} (-\alpha \mathbf{H}_t^{k_i}) = \mathbb{B}_t^{L-1} \mathbb{B}_t^{L-2} \ldots \mathbb{B}_t^0 \mathbf{I}_d. \tag{9}$$

*Proof (sketch).* The proof is carried out by induction on $L$. The base case $L = 1$ is readily shown by applying the definition of $\mathbb{B}_t^i$. Assuming the cases $L = 1, \ldots, L'$ (where $L' < K$) hold, the sums of (9) can be recursively expanded and rewritten in terms of $\mathbb{B}_t^i$ to obtain

$$\mathbf{I}_d + \sum_{l=1}^{L'+1} \sum_{0 \le k_{1:l}\uparrow < K} \prod_{i=1}^{l} (-\alpha \mathbf{H}_t^{k_i}) = \mathbb{B}_t^{L'} \mathbb{B}_t^{L'-1} \ldots \mathbb{B}_t^1 \left[ \mathbf{I}_d + \sum_{k_{L'+1}=k_{L'}+1}^{K-1} (-\alpha \mathbf{H}_t^{k_{L'+1}}) \right]$$

$$= \mathbb{B}_t^{L'} \mathbb{B}_t^{L'-1} \ldots \mathbb{B}_t^1 \mathbb{B}_t^0 \mathbf{I}_d$$

$\square$

Notably, with the definition of matrix product $\mathbf{P}_t^k$, the meta-gradients (4) and (5) can be further simplified as $\nabla \mathcal{L}_t(\boldsymbol{\theta}) = \mathbf{P}_t^K \mathbf{g}_t^K$ and $\hat{\nabla}^{\mathrm{Tr}} \mathcal{L}_t(\boldsymbol{\theta}) = \mathbf{P}_t^L \mathbf{g}_t^K$. Essentially, each matrix $\mathbf{I}_d - \alpha \mathbf{H}_t^k$ represents a mapping $\mathbb{R}^d \mapsto \mathbb{R}^d$ when multiplying with $\mathbf{g}_t^K$, and thus can be viewed as a vector operator in $\mathbb{R}^d$ that requires one HVP computation.

Building upon these insights and Proposition 3.1, we have established the following result.

**Theorem 3.2** (vector operator $\mathbb{B}_t^{\mathbf{g},i}$). *With the notational convention in Proposition 3.1, and given vector $\mathbf{g} \in \mathbb{R}^d$, consider operator $\mathbb{B}_t^{\mathbf{g},i} : \mathbb{R}^d \mapsto \mathbb{R}^d$ such that for any vector $\mathbf{v} \in \mathbb{R}^d$, $\mathbb{B}_t^{\mathbf{g},i} \mathbf{v} := \mathbf{P}_t^i \mathbf{g} - \alpha \sum_{k_{L-i}=k_{L-1-i}+1}^{K-1-i} \mathbf{H}_t^{k_{L-i}} \mathbf{v}$, for $i = 0, 1, \dots, L-1$. For $0 \le L \le K$, it holds that*

$$\hat{\nabla}^{\mathrm{Bi}} \mathcal{L}(\boldsymbol{\theta}) = \mathbb{B}_t^{\mathbf{g}_t^K,L-1} \mathbb{B}_t^{\mathbf{g}_t^K,L-2} \dots \mathbb{B}_t^{\mathbf{g}_t^K,0} \mathbf{g}_t^K. \tag{10}$$

Theorem 3.2 reformulates the complicated binomial expansion in (8) as a series of $L$ vector operators. Note that the vector operator in Theorem 3.2 is distinguished by a superscript from the matrix operator in Proposition 3.1. Similar to the cascaded matrix-vector multiplication in (4) and (5), these vector operators must be carried out from right to left sequentially. However, each operator in (10) may contain multiple computationally independent HVPs that can be readily parallelized on the GPU. The next remark elaborates on this parallelization and analyzes the complexity of the resultant BinomMAML algorithm.

*Remark* 3.3. In Theorem 3.2, each operator requires computing HVPs with $\mathbf{H}_t^{k_{L-i}}$, where the index $k_{L-i} = k_{L-1-i}+1, \dots, K-1-i$. By the recursive relationship of $k_{L-i}$ and the definition $k_0 = -1$, it can be deduced that $k_{L-i}$ ranges from $L-i-1$ to $K-1-i$. Therefore, $K-L+1$ parallel HVPs must be performed per operator, so the overall complexity of (10) is $\mathcal{O}(Ld)$ time and $\mathcal{O}((K-L+1)d)$ space.

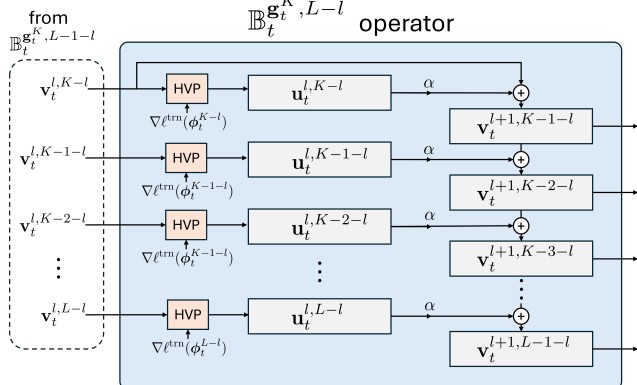

Figure 1: Depiction of $\mathbb{B}_t^{\mathbf{g}_t^K,L-l}$ operator

The Step-by-step pseudocode is presented in Algorithm 1, where each iteration (that is, computing $\{\mathbf{u}_t^{l,k}\}_{k=L-l}^{K-l}$ and $\{\mathbf{v}_t^{l+1,k}\}_{k=L-1-l}^{K-1-l}$) corresponds to one vector operator $\mathbb{B}_t^{\mathbf{g},i}$, with $\{\mathbf{u}_t^{l,k}\}_{k=L-l}^{K-l}$ involving $K-L+1$ parallel HVPs. An illustrative diagram of the $\mathbb{B}_t^{\mathbf{g}_t^K,L-l}$ operator is provided in fig. 1, while a detailed derivation of Algorithm 1 is provided in appendix A.

Complexity considerations bring up two limitations of BinomGBML. First, the parallelization requires $(K - L + 1)$ computational cores compared to a single HVP. While modern GPUs are abundant in such resources, applying BinomGBML to systems lacking cores or without GPUs can be challenging. Second, implementing parallel computations could incur an extra overhead cost. Fortunately, our numerical tests on real datasets suggest this cost is relatively small compared to the HVPs themselves; cf. Figure 4.

Next, we compare the advocated BinomMAML to MAML and its variants outlined in Section 2.2. *Remark* 3.4. First, it is easily seen that BinomMAML conforms with FOMAML when $L = 0$. Moreover, BinomMAML with $L = K$ incurs the same time complexity as MAML, yet a markedly reduced space complexity. This is due to MAML creating the $K$ computation graphs for HVPs when computing $\phi_t^K$, which are saved in memory. In contrast, BinomMAML creates the HVP computational graphs on the fly, and frees up the associated memory once completed. As a side benefit, thus, BinomMAML addresses the space scalability issue in vanilla MAML. For general $L$, BinomMAML and TruncMAML have identical time complexity, but the space complexity of BinomMAML decreases affinely with $L$.

---

**Algorithm 1** BinomMAML's meta-gradient estimation

---

**Input:** training gradients $\{\nabla\ell_t^{\text{trn}}(\phi_t^k)\}_{k=0}^{K-1}$, validation gradient $\mathbf{g}_t^K$,
      step size $\alpha$, truncation $L \in \{1, \ldots, K-1\}$
Initialize $\mathbf{v}_t^{0,k} = \mathbf{g}_t^K$, $k = L, \cdots, K$
**for** $l = 0, \ldots, L-1$ **do**
  $[\mathbf{u}_t^{l,L-l}, \ldots, \mathbf{u}_t^{l,K-l}] = [\nabla_{\phi_t^{L-l}}\langle\nabla\ell^{\text{trn}}(\phi_t^{L-l}), \mathbf{v}_t^{l,L-l}\rangle, \ldots, \nabla_{\phi_t^{K-l}}\langle\nabla\ell^{\text{trn}}(\phi_t^{K-l}), \mathbf{v}_t^{l,K-l}\rangle]$
  $\mathbf{v}_t^{l+1,K-l-1} = \mathbf{v}_t^{l,K-l} - \alpha\mathbf{u}_t^{l,K-l}$
  **for** $k = K-2-l, \ldots, L-1-l$ **do**
    $\mathbf{v}_t^{l+1,k} = \mathbf{v}_t^{l+1,k+1} - \alpha\mathbf{u}_t^{l,k+1}$
  **end for**
**end for**
**Output:** $\hat{\nabla}^{\text{Bi}}\mathcal{L}_t(\boldsymbol{\theta}) = \mathbf{v}_t^{L,0}$

---

## 3.2 ERROR BOUNDS

Having developed the BinomMAML algorithm and elaborated on its computation, this section delves into the estimation errors of FOMAML, TruncMAML, and BinomMAML. The analysis will be performed under three different assumptions that are common and useful in meta-learning.

**Assumption 3.5.** Loss $\ell_t^{\text{trn}}$ has $H$-Lipschitz gradient $\forall t$, i.e., $\|\nabla\ell_t^{\text{trn}}(\phi_t) - \nabla\ell_t^{\text{trn}}(\phi_t')\| \leq H\|\phi_t - \phi_t'\|$, $\forall\phi_t, \phi_t' \in \mathbb{R}^d$.

Assumption 3.5 is very mild and widely used not only for meta-learning (Fallah et al., 2020; Wang et al., 2022), but also more general machine learning (Jain & Kar, 2017; Ji et al., 2021), and can be readily satisfied by a wide range of loss functions and NNs (Virmaux & Scaman, 2018). Under Assumption 3.5, three bounds can be established as follows.

**Theorem 3.6.** *With Assumption 3.5 in effect, it holds that*

$$\|\nabla\mathcal{L}_t(\boldsymbol{\theta}) - \hat{\nabla}^{\text{FO}}\mathcal{L}_t(\boldsymbol{\theta})\| \leq [(1+\alpha H)^K - 1]\|\mathbf{g}_t^K\|, \tag{11a}$$

$$\|\nabla\mathcal{L}_t(\boldsymbol{\theta}) - \hat{\nabla}^{\text{Tr}}\mathcal{L}_t(\boldsymbol{\theta})\| \leq [(1+\alpha H)^K - (1+\alpha H)^L]\|\mathbf{g}_t^K\|, \tag{11b}$$

$$\|\nabla\mathcal{L}_t(\boldsymbol{\theta}) - \hat{\nabla}^{\text{Bi}}\mathcal{L}_t(\boldsymbol{\theta})\| \leq \sum_{l=L+1}^{K}\binom{K}{l}(\alpha H)^l\|\mathbf{g}_t^K\|. \tag{11c}$$

*With $e_t^{\text{FO}}, e_t^{\text{Tr}}, e_t^{\text{Bin}}$ denoting these bounds, it follows that $e_t^{\text{Bin}} < e_t^{\text{Tr}} < e_t^{\text{FO}}$.*

Theorem 3.6 compares the error bounds of the three meta-gradient estimators, where BinomMAML enjoys the smallest loss due to the additional information injected. Note that all three upper bounds are sharp, and can be attained upon setting $\mathbf{H}_t^k = -H\mathbf{I}_d$, $\forall k$. Figure 2a shows the three bounds in Theorem 3.6 across $L$, with $K = 5$, $\alpha = 0.25$, and $H = 1.0$. One can observe that the error of TruncMAML decreases slowly when $L$ is small. In comparison, the error of BinomMAML diminishes swiftly with $L$, and a small error can be readily achieved even with $L = 1$.

Next, a stricter yet important assumption will be considered.

**Assumption 3.7.** The loss function $\ell_t^{\text{trn}}$ is convex $\forall t$.

Although Assumption 3.7 hardly holds for highly non-convex NNs, it is especially useful if the task-specific part of the model is linear; e.g., when adapting solely the last layer of an NN (Revaud et al., 2019; Lee et al., 2019; Oreshkin et al., 2018). This extra assumption on convexity allows us to establish tighter error bounds as follows.

**Theorem 3.8.** *Under Assumptions 3.5 and 3.7, and with $0 < \alpha \leq 1/H$, it holds that*

$$\|\nabla\mathcal{L}_t(\boldsymbol{\theta}) - \hat{\nabla}^{\text{FO}}\mathcal{L}_t(\boldsymbol{\theta})\| \leq [1 - (1-\alpha H)^K]\|\mathbf{g}_t^K\|, \tag{12a}$$

$$\|\nabla\mathcal{L}_t(\boldsymbol{\theta}) - \hat{\nabla}^{\text{Tr}}\mathcal{L}_t(\boldsymbol{\theta})\| \leq [1 - (1-\alpha H)^{K-L}]\|\mathbf{g}_t^K\|, \tag{12b}$$

$$\|\nabla\mathcal{L}_t(\boldsymbol{\theta}) - \hat{\nabla}^{\text{Bi}}\mathcal{L}_t(\boldsymbol{\theta})\| \leq \binom{K}{L+1}(\alpha H)^{L+1}\|\mathbf{g}_t^K\|. \tag{12c}$$

*Moreover, there exists a constant $C_\alpha = \mathcal{O}(K/(L+1))$ such that if $0 < \alpha \leq 1/(C_\alpha H)$, the upper bound in (12c) decreases super-exponentially with $L$.*

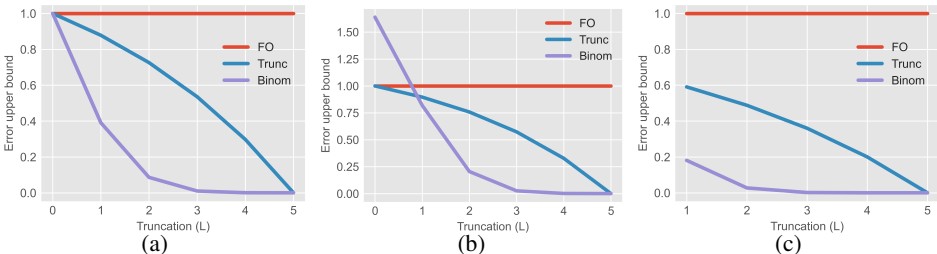

Figure 2: Estimation error upper bounds in Theorems (a) 3.6, (b) 3.8, and (c) 3.10, normalized to FOMAML error.

In addition to convexity, Theorem 3.8 also assumes the learning rate $\alpha$ is not too "aggressive." It is known that a sufficient condition for GD to "descend" is $0 < \alpha < 2/H$, with $\alpha = 1/H$ being an oracle choice (Bertsekas, 2016). Since $H$ is unknown or difficult to estimate in practice, it is common to assume $\alpha = \mathcal{O}(1/H)$ in general, which is also necessary for the analysis of TruncGBML; cf. (Shaban et al., 2019). Estimation error bounds (12) are depicted in Figure 2b. It is important to note that the bounds for FOMAML and TruncMAML are sharp, yet the BinomMAML bound is loose. To be specific, (12a) can be attained by letting $\mathbf{H}_t^k = H\mathbf{I}_d$, $\forall k$, and (12b) is reached when $\mathbf{H}_t^k = H\mathbf{I}_d$, $k = 0, \ldots, K - L - 1$ and $\mathbf{H}_t^k = \mathbf{0}_{d \times d}$, $k = K - L, \ldots, K - 1$. Regardless, the error bound of BinomMAML decreases super-exponentially and outperforms TruncMAML when $L > 0$. Recall that both TruncMAML and BinomMAML are exactly FOMAML with $L = 0$, so their actual errors are identical in this case.

Aside from global convexity, another assumption worth probing is local strong convexity.

**Assumption 3.9.** The loss $\ell_t^{\mathrm{trn}}$ is locally $h$-strongly convex $\forall t$ around $\{\boldsymbol{\phi}_t^k\}_{k=K-M}^{K-1}$, where $M \leq \min\{L, K - L\}$.

Assumption 3.9 is milder than Assumption 3.7 as it presumes strong convexity around only the last $M$ points of the optimization trajectory, which can be sufficiently adjacent to a local optimum. Moreover, such an assumption has been widely used for analyzing bilevel optimization; see e.g., (Ghadimi & Wang, 2018; Shaban et al., 2019).

**Theorem 3.10.** *If Assumptions 3.5 and 3.9 hold, and $0 < \alpha \leq 1/H$, it then follows that*

$$\|\nabla\mathcal{L}_t(\boldsymbol{\theta}) - \hat{\nabla}^{\mathrm{FO}}\mathcal{L}_t(\boldsymbol{\theta})\| \leq \max\{(1 + \alpha H)^{K-M}(1 - \alpha h)^M - 1, 1 - (1 - \alpha H)^K\}\|\mathbf{g}_t^K\|, \quad (13\mathrm{a})$$

$$\|\nabla\mathcal{L}_t(\boldsymbol{\theta}) - \hat{\nabla}^{\mathrm{Tr}}\mathcal{L}_t(\boldsymbol{\theta})\| \leq [(1 + \alpha H)^{K-M} - (1 + \alpha H)^{L-M}](1 - \alpha h)^M\|\mathbf{g}_t^K\|, \quad (13\mathrm{b})$$

$$\|\nabla\mathcal{L}_t(\boldsymbol{\theta}) - \hat{\nabla}^{\mathrm{Bi}}\mathcal{L}_t(\boldsymbol{\theta})\| \leq \left[(\alpha H)^{L+1}\sum_{l=1}^{M}\binom{K-l}{L}(1 - \alpha h)^{l-1}\right. \quad (13\mathrm{c})$$

$$\left. + (1 - \alpha h)^M\sum_{l=L+1}^{K-M}\binom{K-M}{l}(\alpha H)^l\right]\|\mathbf{g}_t^K\|.$$

*Moreover, there exists a constant $C_\alpha' = \mathcal{O}((K-1)/L)$ such that if $0 < \alpha \leq 1/(C_\alpha' H)$, the upper bound in* (13c) *decreases super-exponentially with $L$.*

Again, the estimation error bound of BinomMAML decays more rapidly and is markedly smaller than TruncMAML in this setup, as corroborated by Figure 2c. Since Assumption 3.9 requires $M \leq \min\{L, K - L\}$, the bounds are plotted with $M = 1$, so that $L \in \{1, \ldots, 5\}$. Other parameters are the same as in Figures 2a and 2b, and $h = 0.1$.

## 4 NUMERICAL TESTS

With analytical error bounds established, here we test the empirical performance of BinomMAML on both synthetic and real datasets. Test details can be found in Appendix C.

### 4.1 SYNTHETIC DATA

The first test is a few-shot sinusoid regression problem (Finn et al., 2017), where data are sampled from sinusoids with random phases and amplitudes; each phase-amplitude combination defines a distinct task. Figure 3 compares the actual errors of different meta-gradient estimates calculated on the synthetic sinusoid datasets. Specifically, the left subfigure 3a displays meta-gradient error

Table 1: Few-shot classification accuracies on real datasets with early stopping, where $\pm$ represents sample standard deviation, and the number in parentheses indicates the mean accuracy relative to MAML,  highest one marked in bold.

| L | Method | 5-way miniImageNet (%) | | 5-way tieredImageNet (%) | |
|---|--------|------------|------------|------------|------------|
| | | 1-shot | 5-shot | 1-shot | 5-shot |
| 0 | FOMAML | 44.57 ± 0.92 (-1.93) | 62.97 ± 0.50 (-1.26) | 43.53 ± 0.94 (-3.50) | 63.31 ± 0.52 (-1.81) |
| | Reptile | 42.11 ± 0.92 (-4.39) | 61.07 ± 0.51 (-3.16) | 45.06 ± 0.91 (-1.97) | 61.27 ± 50 (-3.85) |
| 1 | iMAML | 44.47 ± 0.92 (-2.03) | 60.32 ± 0.49 (-3.91) | 44.77 ± 0.98 (-2.26) | 60.81 ± 0.50 (-4.31) |
| | TruncMAML | 44.53 ± 0.89 (-1.97) | **62.43 ± 0.50 (-1.80)** | 43.67 ± 0.96 (-3.36) | 64.15 ± 0.51 (-0.97) |
| | BinomMAML | **45.50 ± 0.91 (-1.00)** | 62.36 ± 0.48 (-1.87) | **46.23 ± 0.95 (-0.80)** | **64.49 ± 0.51 (-0.63)** |
| 2 | iMAML | 44.57 ± 0.95 (-1.93) | 60.63 ± 0.48 (-3.60) | 44.13 ± 0.93 (-2.90) | 61.15 ± 0.50 (-3.97) |
| | TruncMAML | 44.93 ± 0.90 (-1.57) | **63.61 ± 0.48 (-0.62)** | 45.93 ± 0.99 (-1.10) | **64.81 ± 0.49 (-0.31)** |
| | BinomMAML | **46.23 ± 0.92 (-0.27)** | 63.49 ± 0.48 (-0.74) | **46.20 ± 0.92 (-0.83)** | 64.41 ± 0.52 (-0.71) |
| 3 | iMAML | 45.03 ± 0.92 (-1.47) | 62.77 ± 0.48 (-1.46) | 44.80 ± 0.95 (-2.23) | 62.38 ± 0.50 (-2.74) |
| | TruncMAML | 44.33 ± 0.92 (-2.17) | 63.67 ± 0.48 (-0.56) | 45.62 ± 0.98 (-1.41) | 63.69 ± 0.51 (-1.43) |
| | BinomMAML | **46.00 ± 0.92 (-0.50)** | **64.17 ± 0.47 (-0.06)** | **46.43 ± 0.93 (-0.60)** | **65.37 ± 0.49 (+0.25)** |
| 4 | iMAML | 45.43 ± 0.92 (-1.07) | 62.64 ± 0.47 (-1.59) | 45.40 ± 0.95 (-1.63) | 61.80 ± 0.51 (-3.32) |
| | TruncMAML | 44.83 ± 0.93 (-1.67) | 63.44 ± 0.48 (-0.79) | 44.93 ± 0.95 (-2.10) | **64.83 ± 0.51 (-0.29)** |
| | BinomMAML | **46.24 ± 0.94 (-0.16)** | **63.86 ± 0.48 (-0.37)** | **46.73 ± 0.94 (-0.30)** | 64.63 ± 0.50 (-0.49) |
| 5 | MAML | 46.50 ± 0.93 | 64.23 ± 0.50 | 47.03 ± 0.91 | 65.12 ± 0.50 |

averaged on a batch of tasks, where the horizontal axis represents different random batches. With the same truncation $L = 4$, BinomMAML consistently outperforms TruncMAML on varying tasks by an order of $10^3$ to $10^4$. The right subfigure 3b depicts the meta-gradient error across various $L$ values. It is observed that BinomMAML significantly reduces the error relative to TruncMAML. In particular, BinomMAML with $L = 1$ exhibits an error comparable to TruncMAML of $L = 4$, and the error becomes negligible when $L \geq 2$. This suggests BinomMAML can potentially achieve MAML-level performance in certain settings even with a small $L$. These observations corroborate our analysis in Section 3.2. The empirical benefits of BinomMAML are larger than the analytical ones in Fig. 2.

## 4.2 REAL DATA

To assess BinomMAML's performance in practice, real datasets are tested here, including miniImageNet (Vinyals et al., 2016) and tieredImageNet (Ren et al., 2018). The tests follow the "$W$-way $S$-shot" few-shot classification protocol in (Ravi & Larochelle, 2017; Finn et al., 2017), where "way" stands for the number of classes, and "shot" refers to training data per class; that is, $|\mathcal{D}_t^{\text{trn}}| = WS$. The hyperparameters are as in (Finn et al., 2017).

The first test compares the performance of different meta-gradient estimates. To highlight the role of estimation errors, meta-training is *early-stopped* at $20,000$ iterations. Note that MAML can readily converge in such a setup, while the meta-gradient error can slow down convergence when using an estimator. The results are summarized in Table 1, where $L$ represents the truncation parameter of BinomMAML and TruncMAML, and the number of conjugate gradient iterations for iMAML. BinomMAML surpasses TruncMAML in most cases when adopting the same $L$ and exhibits comparable performance for other cases, while outperforming iMAML across the board. Of major note is that BinomMAML dominates TruncMAML (+1.33 average) in the 1-shot regime, while the gap narrows (+0.27 average) in the 5-shot regime. This observation suggests TruncMAML benefits from gradient averaging when data are abundant, whereas BinomMAML remains well-equipped for low-data settings. Moreover, it is seen that the performance gap of BinomMAML relative to vanilla MAML shrinks rapidly with $L$, as opposed to the slow improvement of TruncMAML, and a small $L$ suffices for BinomMAML to attain desirable performance, supporting our analytical results.

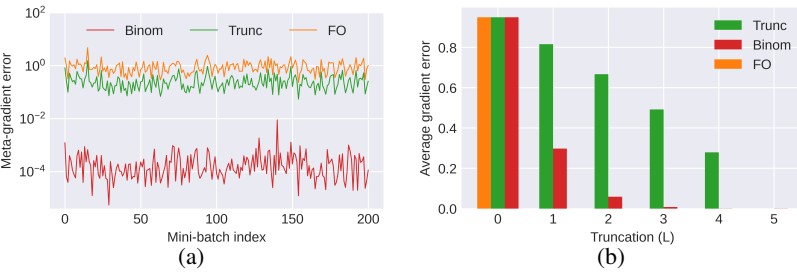

Figure 3: Actual meta-gradient error against (a) different mini-batches of tasks, and (b) truncation $L$.

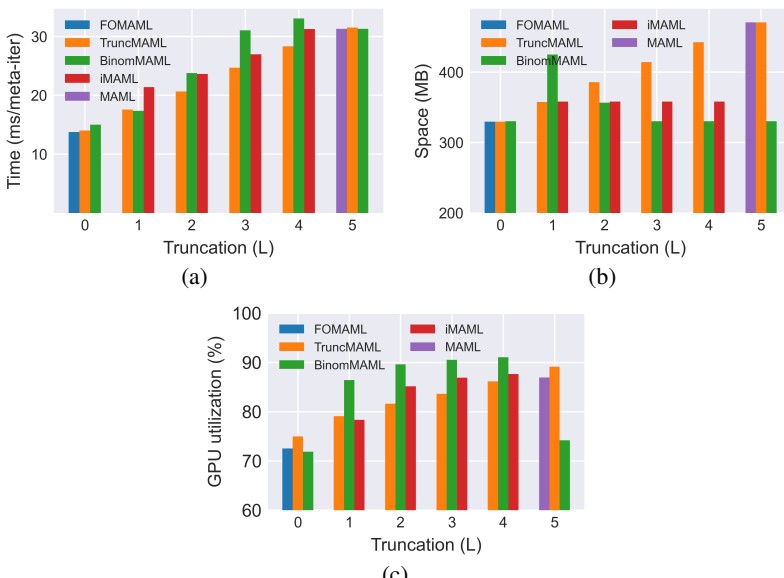

Figure 4: (a) Time complexity; (b) space complexity; and (c) GPU utilization of GBML algorithms on miniImageNet.

The next test showcases the actual time, space, and compute consumption of BinomMAML, as measured under the same miniImageNet test setup. Figure 4 displays the elapsed time per meta-training step, peak GPU memory occupation, and GPU compute core utilization across $L$ values. It should be highlighted that the parallelism, alongside the dynamic creation and release of computation graphs, both incur computational overhead, which varies with $L$. Consequently, the time and compute of BinomMAML slightly outpaces TruncMAML, and the space complexity is not strictly affine in $L$. Further, as no parallelism is required when $L = 0$ or $L = 5$, the time complexity of BinomMAML matches FOMAML and MAML in these two cases. Additionally, BinomMAML leads to markedly reduced memory and compute consumption over vanilla MAML, thanks to the dynamic management of computational graphs; cf. Remark 3.4.

Lastly, Figure 5a depicts the change of accuracy and loss during meta-training, where the curves are smoothed for visualization. Compared to TruncMAML, the convergence of BinomMAML aligns better with MAML. This underscores the importance of reducing meta-gradient error, and in turn explains BinomMAML's superior performance in Table 1.

## 5 CONCLUSIONS

This work delved into the computational complexity of GBML, and developed a reduced-complexity meta-gradient estimator named binomial (Binom)GBML. Inspired by the unique GBML structure in backpropagation, BinomGBML leverages parallelization to maximize the information captured in its meta-gradient estimate. BinomMAML was developed as a running paradigm, showcasing improved error bounds relative to existing approaches. Experiments on synthetic and real datasets corroborate the derived error bounds, which demonstrate the superiority of BinomMAML.

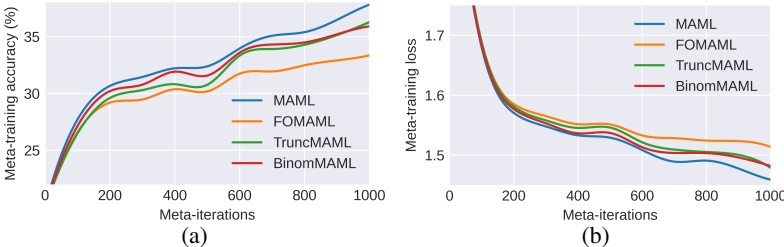

Figure 5: Meta-training (a) accuracy and (b) loss of GBML algorithms on miniImageNet

ETHICS STATEMENT

As this work represents foundational research in meta-learning, we do not anticipate any societal or ethical issues beyond the general and broad-reaching impacts of the advancement of machine learning.

REPRODUCIBILITY STATEMENT

We include as supplementary material the code needed to reproduce the main results on miniImageNet and tieredImageNet. Additionally, all proofs of theoretical results are included in appendix B.

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

## A    BINOMMAML ALGORITHM DERIVATION

**Proposition A.1** (*Proposition 3.1 restated*). *Let* $\mathbf{P}_t^i := \prod_{k=K-i}^{K-1}(\mathbf{I}_d - \alpha\mathbf{H}_t^k)$, $i = 1, \ldots, K$, $\mathbf{P}_t^0 := \mathbf{I}_d$, *and dummy index* $k_0 = -1$. *Define operator* $\mathbb{B}_t^i : \mathbb{R}^{d\times d} \mapsto \mathbb{R}^{d\times d}$ *such that for any matrix* $\mathbf{M} \in \mathbb{R}^{d\times d}$, $\mathbb{B}_t^i\mathbf{M} := \mathbf{P}_t^i - \alpha\sum_{k_{L-i}=k_{L-1-i}+1}^{K-1-i}\mathbf{H}_t^{k_{L-i}}\mathbf{M}$, $i = 0, \ldots, L-1$. *It holds for* $0 \leq L \leq K$ *that*

$$\mathbf{I}_d + \sum_{l=1}^{L}\sum_{0\leq k_{1:l}\uparrow<K}\prod_{i=1}^{l}(-\alpha\mathbf{H}_t^{k_i}) = \mathbb{B}_t^{L-1}\mathbb{B}_t^{L-2}\ldots\mathbb{B}_t^0\mathbf{I}_d \tag{14}$$

*Proof.* The proposition is proved using mathematical induction on $L$. As $L$ will change in this proof, we will alternatively denote $\mathbb{B}_t^i$ as $\mathbb{B}_t^{L,i}$ for notational clarity.

First consider the base case $L = 1$. It can be readily verified from the definition of $\mathbb{B}_t^{L,i}$ that

$$\mathbf{I}_d + \sum_{0\leq k_1<K}(-\alpha\mathbf{H}_t^{k_1}) = \mathbf{P}_t^0 - \alpha\sum_{k_1=k_0+1}^{K-1}\mathbf{H}_t^{k_1} = \mathbb{B}_t^{1,0}\mathbf{I}_d. \tag{15}$$

Now assume (14) holds for $L = 1, \ldots, L'$ where $L' < K$, we next prove that this equation also holds for $L = L' + 1$. It follows that

$$\mathbf{I}_d + \sum_{l=1}^{L'+1}\sum_{0\leq k_1<\ldots<k_l<K}\prod_{i=1}^{l}(-\alpha\mathbf{H}_t^{k_i})$$

$$= \mathbf{I}_d + \sum_{l=1}^{L'}\sum_{0\leq k_1<\ldots<k_l<K}\prod_{i=1}^{l}(-\alpha\mathbf{H}_t^{k_i}) + \sum_{0\leq k_1<\ldots<k_{L'+1}<K}\prod_{i=1}^{l}(-\alpha\mathbf{H}_t^{k_i})$$

$$\overset{(a)}{=} \mathbb{B}_t^{L',L'-1}\mathbb{B}_t^{L',L'-2}\ldots\mathbb{B}_t^{L',0}\mathbf{I}_d + \sum_{0\leq k_1<\ldots<k_{L'+1}<K}\prod_{i=1}^{L'+1}(-\alpha\mathbf{H}_t^{k_i})$$

$$\overset{(b)}{=} \mathbf{P}_t^{L'} - \alpha\sum_{k_1=0}^{K-L'-1}\mathbf{H}_t^{k_1}\mathbb{B}_t^{L',L'-2}\ldots\mathbb{B}_t^{L',0}\mathbf{I}_d + \sum_{k_1=0}^{K-L'-1}\sum_{k_1<k_2<\ldots<k_{L'+1}<K}\prod_{i=1}^{L'+1}(-\alpha\mathbf{H}_t^{k_i})$$

$$= \mathbf{P}_t^{L'} - \alpha\sum_{k_1=0}^{K-L'-1}\mathbf{H}_t^{k_1}\left[\mathbb{B}_t^{L',L'-2}\ldots\mathbb{B}_t^{L',0}\mathbf{I}_d + \sum_{k_1<k_2<\ldots<k_{L'+1}<K}\prod_{i=2}^{L'+1}(-\alpha\mathbf{H}_t^{k_i})\right]$$

$$\overset{(c)}{=} \mathbb{B}_t^{L'+1,L'}\left[\mathbb{B}_t^{L',L'-2}\ldots\mathbb{B}_t^{L',0}\mathbf{I}_d + \sum_{k_1<k_2<\ldots<k_{L'+1}<K}\prod_{i=2}^{L'+1}(-\alpha\mathbf{H}_t^{k_i})\right]$$

$$\overset{(d)}{=} \mathbb{B}_t^{L'+1,L'}\mathbb{B}_t^{L'+1,L'-1}\ldots\mathbb{B}_t^{L'+1,1}\left[\mathbf{I}_d + \sum_{k_{L'+1}=k_{L'}+1}^{K-1}(-\alpha\mathbf{H}_t^{k_{L'+1}})\right]$$

$$= \mathbb{B}_t^{L'+1,L'}\mathbb{B}_t^{L'+1,L'-1}\ldots\mathbb{B}_t^{L'+1,0}\mathbf{I}_d \tag{16}$$

where $(a)$ relies on the inductive hypothesis for $L = L'$, $(b)$ follows from Lemma A.4 and the fact that $\sum_{0\leq k_1<\ldots<k_{L'+1}<K} = \sum_{k_1=0}^{K-L'-1}\sum_{k_1<k_2<\ldots<k_{L'+1}<K}$, $(c)$ is from the definition of $\mathbb{B}_t^{L,i}$, and $(d)$ is by recursively applying $(a)$ through $(c)$. This demonstrates (14) also holds for $L = L' + 1$.

By induction, we conclude that (14) holds for any $L = 1, \ldots, K$, which completes the proof. □

**Theorem A.2** (*Theorem 3.2 restated*). *Let* $\mathbf{P}_t^i := \prod_{k=K-i}^{K-1}(\mathbf{I}_d - \alpha\mathbf{H}_t^k)$, $i = 1, \ldots, K$, $\mathbf{P}_t^0 := \mathbf{I}_d$, *and* $k_0 = -1$. *Given vector* $\mathbf{g} \in \mathbb{R}^d$, *define operator* $\mathbb{B}_t^{\mathbf{g},i} : \mathbb{R}^d \mapsto \mathbb{R}^d$ *such that for any vector* $\mathbf{v} \in \mathbb{R}^d$, $\mathbb{B}_t^{\mathbf{g},i}\mathbf{v} := \mathbf{P}_t^i\mathbf{g} - \alpha\sum_{k_{L-i}=k_{L-1-i}+1}^{K-1-i}\mathbf{H}_t^{k_{L-i}}\mathbf{v}$, $i = 0, \ldots, L-1$. *It holds for* $0 \leq L \leq K$ *that*

$$\hat{\nabla}^{\mathrm{Bi}}\mathcal{L}(\boldsymbol{\theta}) = \mathbb{B}_t^{\mathbf{g}_t^K,L-1}\mathbb{B}_t^{\mathbf{g}_t^K,L-2}\ldots\mathbb{B}_t^{\mathbf{g}_t^K,0}\mathbf{g}_t^K. \tag{17}$$

*Proof.* For any $\mathbf{M} \in \mathbb{R}^{d \times d}$, it can be verified from the definition of $\mathbb{B}_t^i$ that

$$[\mathbb{B}_t^i \mathbf{M}]\mathbf{g} = \left[\mathbf{P}_t^i - \alpha \sum_{k_{L-i}=k_{L-1-i}+1}^{K-1-i} \mathbf{H}_t^{k_{L-i}} \mathbf{M}\right]\mathbf{g} = \mathbf{P}_t^i \mathbf{g} - \alpha \sum_{k_{L-i}=k_{L-1-i}+1}^{K-1-i} \mathbf{H}_t^{k_{L-i}} \mathbf{M}\mathbf{g} = \mathbb{B}_t^{\mathbf{g},i}[\mathbf{M}\mathbf{g}].$$

(18)

Therefore, using Proposition A.1 results in

$$\begin{aligned}
\hat{\nabla}^{\mathrm{Bi}}\mathcal{L}(\boldsymbol{\theta}) &= (\mathbb{B}_t^{L-1}\mathbb{B}_t^{L-2}\ldots\mathbb{B}_t^0\mathbf{I}_d)\mathbf{g}_t^K = [\mathbb{B}_t^{L-1}(\mathbb{B}_t^{L-2}\ldots\mathbb{B}_t^0\mathbf{I}_d)]\mathbf{g}_t^K \\
&\stackrel{(a)}{=} \mathbb{B}_t^{\mathbf{g}_t^K,L-1}[(\mathbb{B}_t^{L-2}\ldots\mathbb{B}_t^0\mathbf{I}_d)\mathbf{g}] \\
&\stackrel{(b)}{=} \mathbb{B}_t^{\mathbf{g}_t^K,L-1}\mathbb{B}_t^{\mathbf{g}_t^K,L-2}\ldots\mathbb{B}_t^{\mathbf{g}_t^K,0}[\mathbf{I}_d\mathbf{g}_t^K] \\
&= \mathbb{B}_t^{\mathbf{g}_t^K,L-1}\mathbb{B}_t^{\mathbf{g}_t^K,L-2}\ldots\mathbb{B}_t^{\mathbf{g}_t^K,0}\mathbf{g}_t^K.
\end{aligned}$$

(19)

where $(a)$ is by applying (18), and $(b)$ is from recursion. $\qquad\square$

*Remark* A.3. As stated in section 3, each $\mathbb{B}_t^{\mathbf{g}_t^K,i}$ corresponds to one iteration of $\mathcal{O}(d)$ parallel HVP computation. Likewise, the meta-gradient calculation in MAML and TruncMAML can be also respectively reformulated as a chain of $K$ and $L$ vector operators.

**Lemma A.4.** *Consider the notation convention in Proposition 3.1. It holds that*

$$\mathbb{B}_t^{L-1}\mathbb{B}_t^{L-2}\ldots\mathbb{B}_t^0\mathbf{I}_d = \mathbf{P}_t^L - \alpha \sum_{k_1=0}^{K-L-1} \mathbf{H}_t^{k_1}\mathbb{B}_t^{L-2}\ldots\mathbb{B}_t^0\mathbf{I}_d.$$

(20)

*Proof.* By the definition of $\mathbb{B}_t^i$, we have

$$\mathbb{B}_t^{L-1}\mathbb{B}_t^{L-2}\ldots\mathbb{B}_t^0\mathbf{I}_d$$

$$= \mathbf{P}_t^{L-1} - \alpha \sum_{k_1=0}^{K-L} \mathbf{H}_t^{k_1}\mathbb{B}_t^{L-2}\ldots\mathbb{B}_t^0\mathbf{I}_d$$

$$\stackrel{(a)}{=} \mathbf{P}_t^{L-1} - \alpha\mathbf{H}_t^{K-L}\left\{\mathbf{P}_t^{L-2} - \alpha \sum_{k_2=K-L+1}^{K-L+1} \mathbf{H}_t^{k_2}\left[\mathbf{P}_t^{L-3} - \ldots (\mathbf{P}_t^0 - \alpha \sum_{k_L=K-1}^{K-1} \mathbf{H}_t^{k_L}\mathbf{I}_d)\right]\right\}$$

$$\quad - \alpha \sum_{k_1=0}^{K-L-1} \mathbf{H}_t^{k_1}\mathbb{B}_t^{L-2}\ldots\mathbb{B}_t^0\mathbf{I}_d$$

$$= \mathbf{P}_t^{L-1} - \alpha\mathbf{H}_t^{K-L}\left\{\mathbf{P}_t^{L-2} - \alpha\mathbf{H}_t^{K-L+1}\left[\mathbf{P}_t^{L-3} - \ldots (\mathbf{P}_t^0 - \alpha\mathbf{H}_t^{K-1})\right]\right\} - \alpha \sum_{k_1=0}^{K-L-1} \mathbf{H}_t^{k_1}\mathbb{B}_t^{L-2}\ldots\mathbb{B}_t^0\mathbf{I}_d$$

$$\stackrel{(b)}{=} \mathbf{P}_t^L - \alpha \sum_{k_1=0}^{K-L-1} \mathbf{H}_t^{k_1}\mathbb{B}_t^{L-2}\ldots\mathbb{B}_t^0\mathbf{I}_d$$

(21)

where $(a)$ separates out the $k_1 = K - L$ term from the summation, and $(b)$ is because $\mathbf{P}_t^i - \alpha\mathbf{H}_t^{K-1-i}\mathbf{P}_t^i = (\mathbf{I}_d - \alpha\mathbf{H}_t^{K-1-i})\mathbf{P}_t^i = \mathbf{P}_t^{i+1}$, $i = 0,\ldots,L-1$ and the definition $\mathbf{P}_t^0 = \mathbf{I}_d$. $\qquad\square$

## A.1 CASE STUDY FOR $L = 2$

To give some intuition on the derivation of the BinomMAML algorithm 2, the BinomMAML expansion (8) truncated to $L = 2$ is examined.

$$\prod_{k=0}^{K-1} [\mathbf{I}_d - \alpha\mathbf{H}_t^k]\mathbf{g}_t^K = \left[\mathbf{I}_d - \alpha \sum_{k_1=0}^{K-1} \mathbf{H}_t^k + \alpha^2 \sum_{k_1=0}^{K-2} \left(\sum_{k_2=k_1+1}^{K-1} \mathbf{H}_t^{k_1}\mathbf{H}_t^{k_2}\right)\right]\mathbf{g}_t^K + o(\alpha^3).$$

The double sum encapsulates all possible combinations of $\mathbf{H}_t^{k_1}\mathbf{H}_t^{k_2}$ from the product expansion. Expand and rewrite

$$\prod_{k=0}^{K-1} [\mathbf{I}_d - \alpha\mathbf{H}_t^k]\mathbf{g}_t^K = \left[\mathbf{I}_d - \alpha\mathbf{H}_t^{K-1} - \alpha \sum_{k_1=0}^{K-2} \mathbf{H}_t^{k_1} + \alpha^2 \sum_{k_1=0}^{K-2} \left(\sum_{k_2=k_1+1}^{K-1} \mathbf{H}_t^{k_1}\mathbf{H}_t^{k_2}\right)\right]\mathbf{g}_t^K$$

$$= \Big[\mathbf{g}_t^K - \alpha\mathbf{H}_t^{K-1}\mathbf{g}_t^K - \alpha\sum_{k_1=0}^{K-2}\mathbf{H}_t^{k_1}\Big(\mathbf{g}_t^K - \alpha\sum_{k_2=k_1+1}^{K-1}\mathbf{H}_t^{k_2}\mathbf{g}_t^K\Big)\Big] \quad (22)$$

How can we calculate (22)? Note for each $k_1$, a unique sum of HVPs over $k_2$ is required. Denote HVPs $\mathbf{u}^{1,k} = \mathbf{H}_t^k\mathbf{g}_t^K, \ k = 1, \ldots, K-1$.

$$\prod_{k=0}^{K-1}[\mathbf{I}_d - \alpha\mathbf{H}_t^k]\mathbf{g}_t^K = \Big[\mathbf{g}_t^K - \alpha\mathbf{H}_t^{K-1}\mathbf{g}_t^K - \alpha\sum_{k_1=0}^{K-2}\mathbf{H}_t^{k_1}\Big(\mathbf{g}_t^K - \alpha\sum_{k_2=k_1+1}^{K-1}\mathbf{u}^{0,k_2}\Big)\Big]$$

Then denote each unique sum as $\mathbf{v}^{1,k} = \mathbf{g}_t^K - \alpha\sum_{k'=k}^{K-1}\mathbf{u}^{0,k'}, \ k = 1, \ldots, K-1$. Letting $\mathbf{v}^{1,K} = \mathbf{g}_t^K$, then this implies $\mathbf{v}^{1,k} = \mathbf{v}^{1,k+1} - \alpha\mathbf{u}^{0,k}, \ k = 1, \ldots, K-1$. Finally, note $\mathbf{g}_t^K - \alpha\mathbf{H}_t^{K-1}\mathbf{g}_t^K = \mathbf{v}^{1,K-1}$.

$$\prod_{k=0}^{K-1}[\mathbf{I}_d - \alpha\mathbf{H}_t^k]\mathbf{g}_t^K = \Big[\mathbf{v}^{1,K-1} - \alpha\sum_{k_1=0}^{K-2}\mathbf{H}_t^{k_1}\mathbf{v}^{1,k_1+1}\Big]$$

The expression is similar to that inside the parentheses above. Indeed, we can define $\mathbf{u}^{1,k} = \mathbf{H}_t^k\mathbf{v}^{1,k+1}, \ k = 0, \ldots, K-2$ and $\mathbf{v}^{2,k} = \mathbf{v}^{1,K-1} - \alpha\sum_{k'=k}^{K-2}\mathbf{u}^{1,k'}$ Rewrite

$$\prod_{k=0}^{K-1}[\mathbf{I}_d - \alpha\mathbf{H}_t^k]\mathbf{g}_t^K = \Big[\mathbf{v}^{1,K-1} - \alpha\sum_{k_1=0}^{K-2}\mathbf{u}^{1,k_1}\Big]$$
$$= \mathbf{v}^{2,0}$$

Which is the final gradient estimation. This procedure boils down to (1) calculate HVPs $\{\mathbf{u}^{0,k}\}_{k=1}^{K-1}$ and sums $\{\mathbf{v}^{1,k}\}_{k=1}^{K-1}$, then (2) calculate HVPs $\{\mathbf{u}^{1,k}\}_{k=0}^{K-2}$ and sums $\{\mathbf{v}^{2,k}\}_{k=0}^{K-2}$. This illustrates some intuition behind the operators $\mathbb{B}_t^{\mathbf{g}_t^K,i}$ introduced in theorem 3.2.

## A.2 Computation of BinomMAML's vector operators

Now consider the more general BinomMAML truncation where $1 \le L \le K$. It will be shown that computing each vector operator $\mathbb{B}_t^{\mathbf{g}_t^K}$ can be done with $K - L + 1$ parallel HVPs in $\mathcal{O}(d)$ time and $\mathcal{O}((K - L + 1)d)$ space. First, denote

$$\mathbf{v}_t^{l,k} = \mathbb{B}_t^{\mathbf{g}_t^K,l-1}\ldots\mathbb{B}_t^{\mathbf{g}_t^K,0}\mathbf{g}_t^K\Big|_{k_{L-l}=k-1} \in \mathbb{R}^d, \quad l = 1, \ldots, L, \quad (23a)$$

$$\mathbf{v}_t^{0,k} = \mathbf{g}_t^K, \ \forall\, k,$$
$$\mathbf{u}_t^{l,k} = \mathbf{H}_t^k\mathbf{v}_t^{l,k} \in \mathbb{R}^d, \quad l = 0, \ldots, L-1. \quad (23b)$$

$\mathbf{v}_t^{l,k}$ an the intermediate term resulting from the chain of operators $\mathbb{B}_t^{\mathbf{g}_t^K,l-1}\ldots\mathbb{B}_t^{\mathbf{g}_t^K,0}\mathbf{g}_t^K$. Since $\mathbb{B}_t^{\mathbf{g}_t^K,l-1}\ldots\mathbb{B}_t^{\mathbf{g}_t^K,0}\mathbf{g}_t^K$ can represent a set of $\mathbf{v}_t^l$ depending on the value of the prior index $k_{L-l}$, the second superscript $k$ dictates $k_{L-l}$. Additionally, $\mathbf{u}_t^{l,k}$ represents an HVP operation whose role is described shortly. With this notation, each $\mathbb{B}_t^{\mathbf{g}_t^K,l}$ is computed using the set of intermediate terms $\{\mathbf{v}_t^{l-1,k+1}\}_{k=L-l+1}^{K-l+1}$ and HVPs $\{\mathbf{u}_t^{l,k}\}_{k=L-l}^{K-l}$. To see how this leads to the BinomMAML algorithm 2, first note that the definitions of $\mathbf{v}_t^{l,k}$ (23a) and $\mathbb{B}_t^{\mathbf{g}_t^K,l}$ (Theorem A.2) lead to

$$\mathbf{v}_t^{l,k} = \mathbb{B}_t^{\mathbf{g}_t^K,l-1}\mathbf{v}_t^{l-1,k+1}\Big|_{k_{L-l}=k-1} = \Big(\mathbf{P}_t^{l-1}\mathbf{g}_t^K - \alpha\sum_{k_{L+1-l}=k_{L-l}+1}^{K-l}\mathbf{H}_t^{k_{L-1-l}}\mathbf{v}_t^{l-1,k_{L+1-l}+1}\Big)\Big|_{k_{L-l}=k-1}, \quad (24a)$$

$$= \Big(\mathbf{P}_t^{l-1}\mathbf{g}_t^K - \alpha\sum_{k_{L+1-l}=k_{L-l}+1}^{K-l}\mathbf{u}_t^{l-1,k_{L+1-l}+1}\Big)\Big|_{k_{L-l}=k-1}. \quad (24b)$$

The above equations expand $\mathbb{B}_t^{\mathbf{g}_t^K,l-1}$ as expressions with $\mathbf{v}_t^{l,k}$ and $\mathbf{u}_t^{l,k}$. In (24a), each $\mathbf{v}_t^{l,k}$ depends on a unique subset of the previously calculated terms $\{\mathbf{v}_t^{l+1,k}\}_{k=L-l-1}^{K-l-1}$, which requires that $\{\mathbf{v}_t^{l,k}\}_{k=L-l}^{K-l}$

---

**Algorithm 2** BinomMAML's meta-gradient estimation (matrix implementation)

---

**Input:** training gradients $\{\nabla\ell_t^{\mathrm{trn}}(\phi_t^k)\}_{k=0}^{K-1}$, validation gradient $\mathbf{g}_t^K$,
    step size $\alpha$, and truncation $L \in \{1, \ldots, K-1\}$
Initialize $\mathbf{V}_t^0 := [\mathbf{g}_t^K, \ldots, \mathbf{g}_t^K] \in \mathbb{R}^{d \times (K-L+1)}$
**for** $l = 0, \ldots, L-1$ **do**
    $\mathbf{U}_t^l = [\nabla_{\phi_t^{L-l-1}}\langle\nabla\ell_t^{\mathrm{trn}}(\phi_t^{L-l-1}), [\mathbf{V}_t^l]_1\rangle, \ldots, \nabla_{\phi_t^{K-l-1}}\langle\nabla\ell_t^{\mathrm{trn}}(\phi_t^{K-l-1}), [\mathbf{V}_t^l]_{K-L+1}\rangle]$
    $\mathbf{V}_t^{l+1} = [\mathbf{V}_t^l]_{K-L+1}\mathbf{1}_{K-L+1}^T - \alpha\mathbf{U}_t^l\mathbf{T}_{K-L+1}$
**end for**
**Output:** $\hat{\nabla}^{\mathrm{Bi}}\mathcal{L}_t(\boldsymbol{\theta}) = [\mathbf{V}_t^L]_1$

---

be saved (rather than summed to find $\mathbb{B}_t^{\mathbf{g}_t^K, l-1}$ directly). (24b) reveals that, though originally unclear, $\mathbf{v}_t^{l,k}$ is simply a sum of HVPs $\{\mathbf{u}_t^{l-1,k}\}_{k=L-l+1}^{K-l+1}$. In fact, (24b) implies that $\mathbf{v}_t^{l,k} = \mathbf{v}_t^{l,k+1} - \alpha\mathbf{u}_t^{l-1,k+1}$, meaning $\{\mathbf{v}_t^{l,k}\}_{k=L-l}^{K-l}$ can be computed as a sequence of computationally trivial additions $\{\mathbf{v}_t^{l,k} = \mathbf{v}_t^{l,k+1} - \alpha\mathbf{u}_t^{l-1,k+1}\}_{k=L-l}^{K-l}$ starting from the highest index $k = K - l$. The $K - L + 1$ computationally independent HVPs can be performed in parallel with $\mathcal{O}(d)$ time and $\mathcal{O}((K-L+1)d)$ space.

Using this framework, BinomMAML (17), as cascade of $L \times \mathbb{B}_t^{\mathbf{g}_t^K, l}$ operators, can be calculated by recursively finding $\{\mathbf{v}_t^{l+1,k}\}_{k=L-l-1}^{K-l-1}$ via the HVPs $\{\mathbf{u}_t^{l,k}\}_{k=L-l}^{K-l}$ for $l = 0, \ldots, L-1$. The total complexity of BinomMAML is then $\mathcal{O}(Ld)$ time and $\mathcal{O}((K-L+1)d)$ space, meaning the space requirement actually decreases affinely with $L$.

The final estimate $\hat{\nabla}^{\mathrm{Bi}}\mathcal{L}(\boldsymbol{\theta})$ is found as

$$\hat{\nabla}^{\mathrm{Bi}}\mathcal{L}(\boldsymbol{\theta}) = \mathbb{B}_t^{\mathbf{g}_t^K, L-1}\mathbb{B}_t^{\mathbf{g}_t^K, L-2}\ldots\mathbb{B}_t^{\mathbf{g}_t^K, 0}\mathbf{v}_t^{0,k}$$
$$= \mathbb{B}_t^{\mathbf{g}_t^K, L-1}\mathbf{v}_t^{L-1,k_1+1}$$
$$= \left[\mathbf{P}_t^{L-1}\mathbf{g}_t^K - \alpha\sum_{k_1=k_0+1}^{K-L}\mathbf{u}_t^{L-1,k_1+1}\right]$$
$$= \mathbf{v}_t^{L,0}.$$

In other words, $\{\mathbf{v}_t^{L,k}\}_{k=0}^{K-L}$ are summed to produce $\hat{\nabla}^{\mathrm{Bi}}\mathcal{L}(\boldsymbol{\theta})$. Finally, it is shown in lemma A.5 that $\mathbf{P}_t^l\mathbf{g}_t^K = \mathbf{v}_t^{l-1,K-l}$, which allows us to reuse $\mathbf{v}_t^{l-1,K-l}$ as $\mathbf{P}_t^l\mathbf{g}_t^K$ rather than directly calculate $\mathbf{P}_t^l\mathbf{g}_t^K = (\mathbf{I}_d - \alpha\mathbf{H}_t^{K-l})\mathbf{P}_t^{l-1}\mathbf{g}_t^K$. This recursive operator structure leads to Algorithm 1, requiring $\mathcal{O}(Ld)$ time and $\mathcal{O}((K-L+1)d)$ space.

However, Algorithm 1 can be improved by reformulating the vector additions in the inner *for* loop as a matrix multiplication to take advantage of low-level matrix optimizations in libraries such as PyTorch. This can be achieved by stacking $\{\mathbf{v}_t^{l-1,k+1}\}_{k=L-l+1}^{K-l+1}$ and $\{\mathbf{u}_t^{l,k}\}_{k=L-l}^{K-l}$ as the columns of the matrices

$$\mathbf{V}_t^l := [\mathbf{v}_t^{l,L-l}, \ldots, \mathbf{v}_t^{l,K-l}],$$
$$\mathbf{U}_t^l := [\mathbf{u}_t^{l,L-l}, \ldots, \mathbf{u}_t^{l,K-l}].$$

Then it follows that

$$\mathbf{V}_t^{l+1} = \mathbf{v}_t^{l,K-l-1}\mathbf{1}_{K-L+1}^T - \alpha\mathbf{U}_t^l\mathbf{T}_{K-L+1},$$

where $\mathbf{T}_{K-L+1} \in \mathbb{R}^{(K-L+1)\times(K-L+1)}$ is a strictly lower-triangular matrix filled with 1's. This reformulation produces Algorithm 2.

**Lemma A.5.** *Consider the notation $\mathbf{v}_t^{l,k}$ in (23a). It holds for $1 \le l < L$ that*

$$\mathbf{P}_t^l\mathbf{g}_t^K = \mathbf{v}_t^{l-1,K-l}. \tag{25}$$

*Proof.* As a result of lemma A.4, we have the two equations

$$\mathbb{B}_t^{\mathbf{g}_t^K,l-1}\mathbb{B}_t^{\mathbf{g}_t^K,l-2}\ldots\mathbb{B}_t^{\mathbf{g}_t^K,0}\mathbf{g}_t^K = \mathbf{P}_t^{l-1}\mathbf{g}_t^K - \alpha\sum_{k_{L+1-l}=k_{L-l}+1}^{K-l}\mathbf{H}_t^{k_{L+1-l}}\mathbb{B}_t^{\mathbf{g}_t^K,l-2}\ldots\mathbb{B}_t^{\mathbf{g}_t^K,0}\mathbf{g}_t^K,$$

(26a)

$$\mathbb{B}_t^{\mathbf{g}_t^K,l-1}\mathbb{B}_t^{\mathbf{g}_t^K,l-2}\ldots\mathbb{B}_t^{\mathbf{g}_t^K,0}\mathbf{g}_t^K = \mathbf{P}_t^{l}\mathbf{g}_t^K - \alpha\sum_{k_{L+1-l}=k_{L-l}+1}^{K-1-l}\mathbf{H}_t^{k_{L+1-l}}\mathbb{B}_t^{\mathbf{g}_t^K,l-2}\ldots\mathbb{B}_t^{\mathbf{g}_t^K,0}\mathbf{g}_t^K.$$ (26b)

Subtracting (26b) from (26a), we get

$$\mathbf{P}_t^l\mathbf{g}_t^K = \mathbf{P}_t^{l-1}\mathbf{g}_t^K - \alpha\mathbf{H}_t^{K-l}\mathbb{B}_t^{\mathbf{g}_t^K,l-2}\ldots\mathbb{B}_t^{\mathbf{g}_t^K,0}\mathbf{g}_t^K\Big|_{k_{L-l}=K-1-l}$$

Or, by the definition of $\mathbb{B}_t^{\mathbf{g}_t^K,i}$, that

$$\mathbf{P}_t^l\mathbf{g}_t^K = \mathbb{B}_t^{\mathbf{g}_t^K,l-1}\mathbb{B}_t^{\mathbf{g}_t^K,l-2}\ldots\mathbb{B}_t^{\mathbf{g}_t^K,0}\mathbf{g}_t^K\Big|_{k_{L-l}=K-1-l} = \mathbf{v}_t^{l,K-l}$$

$\square$

## B  ERROR ANALYSIS

**Theorem B.1** (*Theorem 3.6 restated*). *With Assumption 3.5 in effect, it holds that*

$$\|\nabla\mathcal{L}_t(\boldsymbol{\theta}) - \hat{\nabla}^{\mathrm{FO}}\mathcal{L}_t(\boldsymbol{\theta})\| \leq [(1+\alpha H)^K - 1]\|\mathbf{g}_t^K\|,$$ (27a)

$$\|\nabla\mathcal{L}_t(\boldsymbol{\theta}) - \hat{\nabla}^{\mathrm{Tr}}\mathcal{L}_t(\boldsymbol{\theta})\| \leq [(1+\alpha H)^K - (1+\alpha H)^L]\|\mathbf{g}_t^K\|,$$ (27b)

$$\|\nabla\mathcal{L}_t(\boldsymbol{\theta}) - \hat{\nabla}^{\mathrm{Bi}}\mathcal{L}_t(\boldsymbol{\theta})\| \leq \left[\sum_{l=L+1}^{K}\binom{K}{l}(\alpha H)^l\right]\|\mathbf{g}_t^K\|.$$ (27c)

*Moreover, denoting these three upper bounds by $e_t^{\mathrm{FO}}, e_t^{\mathrm{Tr}}, e_t^{\mathrm{Bin}}$, it follows that $e_t^{\mathrm{Bin}} < e_t^{\mathrm{Tr}} < e_t^{\mathrm{FO}}$.*

*Proof.* Using Lemma B.2 yields

$$\|\nabla\mathcal{L}_t(\boldsymbol{\theta}) - \hat{\nabla}^{\mathrm{FO}}\mathcal{L}_t(\boldsymbol{\theta})\| = \|\mathbf{P}_t^K\mathbf{g}_t^K - \mathbf{g}_t^K\| \leq \|\mathbf{P}_t^K - \mathbf{I}_d\|\|\mathbf{g}_t^K\| \leq [(1+\alpha H)^K - 1]\|\mathbf{g}_t^K\|.$$

For TruncMAML, we have that

$$\|\nabla\mathcal{L}_t(\boldsymbol{\theta}) - \hat{\nabla}^{\mathrm{Tr}}\mathcal{L}_t(\boldsymbol{\theta})\| = \|\mathbf{P}_t^K\mathbf{g}_t^K - \mathbf{P}_t^L\mathbf{g}_t^K\| \leq \left\|\prod_{k=0}^{K-L-1}(\mathbf{I}_d - \alpha\mathbf{H}_t^k) - \mathbf{I}_d\right\|\|\mathbf{P}_t^L\|\|\mathbf{g}_t^K\|$$ (28)

$$\leq [(1+\alpha H)^{K-L} - 1](1+\alpha H)^L\|\mathbf{g}_t^K\|$$

where the last line utilizes Lemma B.2 and that $\|\mathbf{P}_t^L\| \leq \prod_{k=K-L}^{K-1}\|1 - \alpha\mathbf{H}_t^k\| \leq (1+\alpha H)^L$.

Next, we prove the upper bound for BinomMAML. Notice from (8) that

$$\|\nabla\mathcal{L}_t(\boldsymbol{\theta}) - \hat{\nabla}^{\mathrm{Bi}}\mathcal{L}_t(\boldsymbol{\theta})\| \leq \left\|\mathbf{P}_t^K - \left[\mathbf{I}_d + \sum_{l=1}^{L}\sum_{0\leq k_1<\ldots<k_l<K}\prod_{i=1}^{l}(-\alpha\mathbf{H}_t^{k_i})\right]\right\|\|\mathbf{g}_t^K\|$$

$$\overset{(a)}{=} \left\|\mathbf{I}_d - \sum_{l=1}^{K}\sum_{0\leq k_1<\ldots<k_l<K}\prod_{i=1}^{l}(-\alpha\mathbf{H}_t^{k_i}) - \left[\mathbf{I}_d + \sum_{l=1}^{L}\sum_{0\leq k_1<\ldots<k_l<K}\prod_{i=1}^{l}(-\alpha\mathbf{H}_t^{k_i})\right]\right\|\|\mathbf{g}_t^K\|$$

$$= \left\|\sum_{l=L+1}^{K}\sum_{0\leq k_1<\ldots<k_l<K}\prod_{i=1}^{l}(-\alpha\mathbf{H}_t^{k_i})\right\|\|\mathbf{g}_t^K\|$$ (29)

$$\leq \sum_{l=L+1}^{K}\sum_{0\leq k_1<\ldots<k_l<K}\prod_{i=1}^{l}\alpha\|\mathbf{H}_t^{k_i}\|\|\mathbf{g}_t^K\|$$

$$\overset{(b)}{\leq} \sum_{l=L+1}^{K} \binom{K}{l} (\alpha H)^l \|\mathbf{g}_t^K\|$$

where $(a)$ applies binomial theorem to $\mathbf{P}_t^K = \prod_{k=0}^{K-1}(\mathbf{I}_d - \alpha \mathbf{H}_t^k)$, and $(b)$ uses Assumption 3.5.

Lastly, we prove that $e_t^{\mathrm{Bin}} < e_t^{\mathrm{Tr}} < e_t^{\mathrm{FO}}$. As $(1 + \alpha H)^L > 1$ for $1 \leq L < K$, it is easy to see that

$$e_t^{\mathrm{Tr}} = [(1 + \alpha H)^K - (1 + \alpha H)^L]\|\mathbf{g}_t^K\| < [(1 + \alpha H)^K - 1]\|\mathbf{g}_t^K\| = e_t^{\mathrm{FO}}.$$

In addition, we have

$$e_t^{\mathrm{Tr}} - e_t^{\mathrm{Bin}} = \left[ (1 + \alpha H)^K - (1 + \alpha H)^L - \sum_{l=L+1}^{K} \binom{K}{l}(\alpha H)^l \right] \|\mathbf{g}_t^K\|$$

$$\overset{(a)}{=} \left[ \sum_{l=0}^{K} \binom{K}{l}(\alpha H)^l - \sum_{l=0}^{L} \binom{L}{l}(\alpha H)^l - \sum_{l=L+1}^{K} \binom{K}{l}(\alpha H)^l \right] \|\mathbf{g}_t^K\|$$

$$= \left[ \sum_{l=0}^{L} \binom{K}{l}(\alpha H)^l - \sum_{l=0}^{L} \binom{L}{l}(\alpha H)^l \right] \|\mathbf{g}_t^K\|$$

$$= \left\{ \sum_{l=0}^{L} \left[ \binom{K}{l} - \binom{L}{l} \right](\alpha H)^l \right\} \|\mathbf{g}_t^K\| \overset{(b)}{>} 0$$

where $(a)$ relies on the binomial theorem, and $(b)$ is due to $\binom{K}{l} - \binom{L}{l} > 0$ for $K > L$. The proof is thus completed. $\qquad\square$

**Lemma B.2.** *With Assumption 3.5 in effect, it holds for $i = 0, \dots, K$ that*

$$\|\mathbf{P}_t^i - \mathbf{I}_d\| \leq (1 + \alpha H)^i - 1. \tag{30}$$

*Proof.* First notice that

$$\mathbf{P}_t^i - \mathbf{I}_d = (\mathbf{I}_d - \alpha \mathbf{H}_t^{K-i})\mathbf{P}_t^{i-1} - \mathbf{I}_d$$

$$= (\mathbf{P}_t^{i-1} - \mathbf{I}_d) - \alpha \mathbf{H}_t^{K-i}\mathbf{P}_t^{i-1}$$

$$\overset{(a)}{=} (\mathbf{P}_t^0 - \mathbf{I}_d) - \alpha \sum_{k=K-i}^{K-1} \mathbf{H}_t^k \mathbf{P}_t^{K-k-1}$$

$$= -\alpha \sum_{k=K-i}^{K-1} \mathbf{H}_t^k \mathbf{P}_t^{K-k-1}$$

where $(a)$ is by telescoping the second line.

Then, it follows from Assumption 3.5 and the definition of $\mathbf{P}_t^i$ that

$$\|\mathbf{P}_t^i - \mathbf{I}_d\| = \alpha \left\| \sum_{k=K-i}^{K-1} \mathbf{H}_t^k \mathbf{P}_t^{K-1-k} \right\|$$

$$\leq \alpha \sum_{k=K-i}^{K-1} \|\mathbf{H}_t^k\| \|\mathbf{P}_t^{K-1-k}\|$$

$$\leq \alpha H \sum_{k=K-i}^{K-1} (1 + \alpha H)^{K-1-k}$$

$$= (1 + \alpha H)^i - 1$$

which completes the proof. $\qquad\square$

**Theorem B.3** (*Theorem 3.8 restated*). *Assume Assumptions 3.5 and 3.7 hold, and $0 < \alpha \le 1/H$. It follows that*

$$\|\nabla\mathcal{L}_t(\boldsymbol{\theta}) - \hat{\nabla}^{\mathrm{FO}}\mathcal{L}_t(\boldsymbol{\theta})\| \le [1 - (1 - \alpha H)^K]\|\mathbf{g}_t^K\|, \tag{31a}$$

$$\|\nabla\mathcal{L}_t(\boldsymbol{\theta}) - \hat{\nabla}^{\mathrm{Tr}}\mathcal{L}_t(\boldsymbol{\theta})\| \le [1 - (1 - \alpha H)^{K-L}]\|\mathbf{g}_t^K\|, \tag{31b}$$

$$\|\nabla\mathcal{L}_t(\boldsymbol{\theta}) - \hat{\nabla}^{\mathrm{Bi}}\mathcal{L}_t(\boldsymbol{\theta})\| \le \binom{K}{L+1}(\alpha H)^{L+1}\|\mathbf{g}_t^K\|. \tag{31c}$$

*Moreover, there exists a constant $C_\alpha = \mathcal{O}(K/(L+1))$ such that if $0 < \alpha \le 1/(C_\alpha H)$, the upper bound in* (31c) *decreases super-exponentially with $L$.*

*Proof.* By Assumption 3.7 and $0 < \alpha H \le 1$, we have $0 \le 1 - \alpha H \preceq \mathbf{I}_d - \alpha\mathbf{H}_t^k \preceq 1$. It follows that $\|\mathbf{P}_t^i\| \le \prod_{k=K-i}^{K-1}\|\mathbf{I}_d - \alpha\mathbf{H}_t^k\| \le 1$, and the smallest eigenvalue $\lambda_{\min}(\mathbf{P}_t^i) \ge \prod_{k=K-i}^{K-1}\lambda_{\min}(\mathbf{I}_d - \alpha\mathbf{H}_t^k) = (1 - \alpha H)^i \ge 0$. As a consequence, we obtain $0 \preceq \mathbf{I}_d - \mathbf{P}_t^i \preceq 1 - (1 - \alpha H)^i$, thus giving

$$\|\nabla\mathcal{L}_t(\boldsymbol{\theta}) - \hat{\nabla}^{\mathrm{FO}}\mathcal{L}_t(\boldsymbol{\theta})\| \le \|\mathbf{P}_t^K - \mathbf{I}_d\|\|\mathbf{g}_t^K\| \le [1 - (1 - \alpha H)^K]\|\mathbf{g}_t^K\|.$$

Similarly, we have for TruncMAML that

$$\|\nabla\mathcal{L}_t(\boldsymbol{\theta}) - \hat{\nabla}^{\mathrm{Tr}}\mathcal{L}_t(\boldsymbol{\theta})\| \le \left\|\prod_{k=0}^{K-L-1}(\mathbf{I}_d - \alpha\mathbf{H}_t^k) - \mathbf{I}_d\right\|\|\mathbf{P}_t^L\|\|\mathbf{g}_t^K\| \le [1 - (1 - \alpha H)^{K-L}]\|\mathbf{g}_t^K\|.$$

Regarding BinomMAML, using Lemma B.4 gives

$$
\begin{aligned}
\|\nabla\mathcal{L}_t(\boldsymbol{\theta}) - \hat{\nabla}^{\mathrm{Bi}}\mathcal{L}_t(\boldsymbol{\theta})\| &= \left\|\sum_{l=1}^{K-L}\left[\sum_{0 \le k_1 < \ldots < k_L \le K-1-l}\prod_{i=1}^{L}(-\alpha\mathbf{H}_t^{k_i})\right](-\alpha\mathbf{H}_t^{K-l})\mathbf{P}_t^{l-1}\mathbf{g}_t^K\right\| \\
&\le \sum_{l=1}^{K-L}\left[\sum_{0 \le k_1 < \ldots < k_L \le K-1-l}\prod_{i=1}^{L}\alpha\|\mathbf{H}_t^{k_i}\|\right]\alpha\|\mathbf{H}_t^{K-l}\|\|\mathbf{P}_t^{l-1}\|\|\mathbf{g}_t^K\| \\
&\overset{(a)}{\le} \left[\sum_{l=1}^{K-L}\binom{K-l}{L}\right](\alpha H)^{L+1}\|\mathbf{g}_t^K\|
\end{aligned}
$$

where $(a)$ relies on Assumption 3.5 and $\|\mathbf{P}_t^i\| \le 1$.

To obtain the desired result (27c), notice that

$$
\begin{aligned}
\sum_{l=1}^{K-L}\binom{K-l}{L} &= \binom{L}{L} + \binom{L+1}{L} + \sum_{l=1}^{K-L-2}\binom{K-L}{L} \\
&\overset{(a)}{=} \binom{L+1}{L+1} + \binom{L+1}{L} + \sum_{l=1}^{K-L-2}\binom{K-L}{L} \\
&\overset{(b)}{=} \binom{L+2}{L+1} + \sum_{l=1}^{K-L-2}\binom{K-L}{L} \\
&\overset{(c)}{=} \binom{K}{L+1}
\end{aligned}
$$

where $(a)$ uses $\binom{L}{L} = 1 = \binom{L+1}{L+1}$, $(b)$ is because $\binom{L+1}{k+1} + \binom{L+1}{k} = \binom{L+2}{k+1}$ for $0 \le k \le L$, and $(c)$ is by recursively applying $(a)$ through $(b)$.

Lastly, using Lemma B.5 incurs

$$\|\nabla\mathcal{L}_t(\boldsymbol{\theta}) - \hat{\nabla}^{\mathrm{Bi}}\mathcal{L}_t(\boldsymbol{\theta})\| \le \binom{K}{L+1}(\alpha H)^{L+1}\|\mathbf{g}_t^K\| < \left(\alpha H\frac{eK}{L+1}\right)^{L+1}\|\mathbf{g}_t^K\|.$$

The second bound is not tight. As a result, there exists $C_\alpha \le eK/(L+1)$ to ensure the super-exponential decrease wrt $L$, thus completing the proof. $\qquad\square$

**Lemma B.4.** *It holds for $1 \leq L < K$ that*

$$\nabla \mathcal{L}_t(\boldsymbol{\theta}) - \hat{\nabla}^{\mathrm{Bi}} \mathcal{L}_t(\boldsymbol{\theta}) = \sum_{l=1}^{K-L} \left[ \sum_{0 \leq k_1 < \ldots < k_L \leq K-1-l} \prod_{i=1}^{l} (-\alpha \mathbf{H}_t^{k_i}) \right] (-\alpha \mathbf{H}_t^{K-l}) \mathbf{P}_t^{l-1} \mathbf{g}_t^K.$$

*Proof.* Using binomial theorem as in (29) yields

$$\nabla \mathcal{L}_t(\boldsymbol{\theta}) - \hat{\nabla}^{\mathrm{Bi}} \mathcal{L}_t(\boldsymbol{\theta}) = \sum_{l=L+1}^{K} \sum_{0 \leq k_1 < \ldots < k_l \leq K-1} \prod_{i=1}^{l} (-\alpha \mathbf{H}_t^{k_i}) \mathbf{g}_t^K := \mathbf{M}_t[K] \mathbf{g}_t^K$$

$$\overset{(a)}{=} \left[ \sum_{l=L+1}^{K} \sum_{0 \leq k_1 < \ldots < k_{l-1} \leq K-2} \prod_{i=1}^{l-1} (-\alpha \mathbf{H}_t^{k_i}) \right] (-\alpha \mathbf{H}_t^{K-1}) \mathbf{g}_t^K + \sum_{l=L+1}^{K-1} \sum_{0 \leq k_1 < \ldots < k_l \leq K-2} \prod_{i=1}^{l} (-\alpha \mathbf{H}_t^{k_i}) \mathbf{g}_t^K$$

$$\overset{(b)}{=} \left[ \sum_{l=L}^{K-1} \sum_{0 \leq k_1 < \ldots < k_l \leq K-2} \prod_{i=1}^{l} (-\alpha \mathbf{H}_t^{k_i}) \right] (-\alpha \mathbf{H}_t^{K-1}) \mathbf{g}_t^K + \sum_{l=L+1}^{K-1} \sum_{0 \leq k_1 < \ldots < k_l \leq K-2} \prod_{i=1}^{l} (-\alpha \mathbf{H}_t^{k_i}) \mathbf{g}_t^K$$

$$\overset{(c)}{=} \left[ \sum_{0 \leq k_1 < \ldots < k_L \leq K-2} \prod_{i=1}^{L} (-\alpha \mathbf{H}_t^{k_i}) \right] (-\alpha \mathbf{H}_t^{K-1}) \mathbf{g}_t^K + \left[ \sum_{l=L+1}^{K-1} \sum_{0 \leq k_1 < \ldots < k_l \leq K-2} \prod_{i=1}^{l} (-\alpha \mathbf{H}_t^{k_i}) \right] (-\alpha \mathbf{H}_t^{K-1}) \mathbf{g}_t^K +$$

$$\left[ \sum_{l=L+1}^{K-1} \sum_{0 \leq k_1 < \ldots < k_l \leq K-2} \prod_{i=1}^{l} (-\alpha \mathbf{H}_t^{k_i}) \right] \mathbf{g}_t^K$$

$$= \left[ \sum_{0 \leq k_1 < \ldots < k_L \leq K-2} \prod_{i=1}^{L} (-\alpha \mathbf{H}_t^{k_i}) \right] (-\alpha \mathbf{H}_t^{K-1}) \mathbf{g}_t^K + \left[ \sum_{l=L+1}^{K-1} \sum_{0 \leq k_1 < \ldots < k_l \leq K-2} \prod_{i=1}^{l} (-\alpha \mathbf{H}_t^{k_i}) \right] (\mathbf{I}_d - \alpha \mathbf{H}_t^{K-1}) \mathbf{g}_t^K$$

$$= \left[ \sum_{0 \leq k_1 < \ldots < k_L \leq K-2} \prod_{i=1}^{L} (-\alpha \mathbf{H}_t^{k_i}) \right] (-\alpha \mathbf{H}_t^{K-1}) \mathbf{g}_t^K + \mathbf{M}_t[K-1] (\mathbf{I}_d - \alpha \mathbf{H}_t^{K-1}) \mathbf{g}_t^K \qquad (32)$$

where $(a)$ splits the summation into two parts based on whether $k_l = K - 1$, $(b)$ rewrites the first summation by replacing index $l$ with $l + 1$, and $(c)$ separates out the $l = L$ term from the first summation.

Telescoping the series $\mathbf{M}_t[K]$ in (32) until $\mathbf{M}_t[L+1]$ leads to

$$\nabla \mathcal{L}_t(\boldsymbol{\theta}) - \hat{\nabla}^{\mathrm{Bi}} \mathcal{L}_t(\boldsymbol{\theta})$$

$$= \sum_{l=1}^{K-1-L} \left[ \sum_{0 \leq k_1 < \ldots < k_L \leq K-1-l} \prod_{i=1}^{L} (-\alpha \mathbf{H}_t^{k_i}) \right] (-\alpha \mathbf{H}_t^{K-l}) \mathbf{P}_t^{l-1} \mathbf{g}_t^K + \mathbf{M}_t[L+1] \mathbf{P}_t^{K-1-L} \mathbf{g}_t^K$$

$$= \sum_{l=1}^{K-1-L} \left[ \sum_{0 \leq k_1 < \ldots < k_L \leq K-1-l} \prod_{i=1}^{L} (-\alpha \mathbf{H}_t^{k_i}) \right] (-\alpha \mathbf{H}_t^{K-l}) \mathbf{P}_t^{l-1} \mathbf{g}_t^K + \left[ \prod_{k=0}^{L} (-\alpha \mathbf{H}_t^k) \right] \mathbf{P}_t^{K-1-L} \mathbf{g}_t^K$$

$$= \sum_{l=1}^{K-L} \left[ \sum_{0 \leq k_1 < \ldots < k_L \leq K-1-l} \prod_{i=1}^{L} (-\alpha \mathbf{H}_t^{k_i}) \right] (-\alpha \mathbf{H}_t^{K-l}) \mathbf{P}_t^{l-1} \mathbf{g}_t^K \qquad (33)$$

which is the sought result. $\qquad \square$

**Lemma B.5** ((Cormen et al., 2022)). *For $L = 0, \ldots, K$, it holds*

$$\binom{K}{L} < \left( \frac{eK}{L} \right)^L$$

*where $e$ is Euler's number.*

**Theorem B.6** (*Theorem 3.10 restated*)**.** *Assume Assumptions 3.5 and 3.9 hold, and $0 < \alpha \leq 1/H$. It follows that*

$$\|\nabla \mathcal{L}_t(\boldsymbol{\theta}) - \hat{\nabla}^{\mathrm{FO}} \mathcal{L}_t(\boldsymbol{\theta})\| \leq \max\{(1 + \alpha H)^{K-M}(1 - \alpha h)^M - 1, 1 - (1 - \alpha H)^K\} \|\mathbf{g}_t^K\|, \quad (34a)$$

$$\|\nabla\mathcal{L}_t(\boldsymbol{\theta}) - \hat{\nabla}^{\mathrm{Tr}}\mathcal{L}_t(\boldsymbol{\theta})\| \leq [(1+\alpha H)^{K-M} - (1+\alpha H)^{L-M}](1-\alpha h)^M \|\mathbf{g}_t^K\|, \tag{34b}$$

$$\|\nabla\mathcal{L}_t(\boldsymbol{\theta}) - \hat{\nabla}^{\mathrm{Bi}}\mathcal{L}_t(\boldsymbol{\theta})\| \leq$$
$$\left[(\alpha H)^{L+1}\sum_{l=1}^{M}\binom{K-l}{L}(1-\alpha h)^{l-1} + (1-\alpha h)^M \sum_{l=L+1}^{K-M}\binom{K-M}{l}(\alpha H)^l\right]\|\mathbf{g}_t^K\|. \tag{34c}$$

*Moreover, there exists a constant $C_\alpha' = \mathcal{O}((K-1)/L)$ such that if $0 < \alpha \leq 1/(C_\alpha H)$, the upper bound in (34c) decreases super-exponentially with $L$.*

*Proof.* Assumption 3.9 implies for $k = K-M,\ldots,K-1$, it holds $h \preceq \mathbf{H}_t^k \preceq H$, and hence $0 < 1-\alpha H \preceq \mathbf{I}_d - \alpha\mathbf{H}_t^k \preceq 1-\alpha h$. In addition, $0 < \alpha H \leq 1$ leads to $0 \leq 1-\alpha H \preceq \mathbf{I}_d - \alpha\mathbf{H}_t^k \preceq 1+\alpha H$, $k = 0,\ldots,K-M-1$. Therefore, $\mathbf{P}_t^K \succeq 0$, and the error of FO-MAML has upper bound

$$\|\nabla\mathcal{L}_t(\boldsymbol{\theta}) - \hat{\nabla}^{\mathrm{FO}}\mathcal{L}_t(\boldsymbol{\theta})\| \leq \|\mathbf{P}_t^K - \mathbf{I}_d\|\|\mathbf{g}_t^K\| = \left\|\prod_{k=0}^{K-1-M}(\mathbf{I}_d - \alpha\mathbf{H}_t^k)\prod_{k=K-M}^{K-1}(\mathbf{I}_d - \alpha\mathbf{H}_t^k) - \mathbf{I}_d\right\|\|\mathbf{g}_t^K\|$$
$$\leq \max\{(1+\alpha H)^{K-M}(1-\alpha h)^M - 1, 1 - (1-\alpha H)^K\}\|\mathbf{g}_t^K\|.$$

Moreover, it follows from (28) that

$$\|\nabla\mathcal{L}_t(\boldsymbol{\theta}) - \hat{\nabla}^{\mathrm{Tr}}\mathcal{L}_t(\boldsymbol{\theta})\| \leq \left\|\prod_{k=0}^{K-L-1}(\mathbf{I}_d - \alpha\mathbf{H}_t^k) - \mathbf{I}_d\right\|\|\mathbf{P}_t^L\|\|\mathbf{g}_t^K\|$$
$$\leq [(1+\alpha H)^{K-L} - 1](1+\alpha H)^{L-M}(1-\alpha h)^M\|\mathbf{g}_t^K\|.$$

Next, we prove the upper bound for our BinomMAML. Telescoping the series $\mathbf{M}_t[K]$ in (32) through $\mathbf{M}_t[K-M]$ leads to

$$\nabla\mathcal{L}_t(\boldsymbol{\theta}) - \hat{\nabla}^{\mathrm{Bi}}\mathcal{L}_t(\boldsymbol{\theta}) \tag{35}$$
$$= \sum_{l=1}^{M}\left[\sum_{0\leq k_1<\ldots<k_L\leq K-1-l}\prod_{i=1}^{L}(-\alpha\mathbf{H}_t^{k_i})\right](-\alpha\mathbf{H}_t^{K-l})\mathbf{P}_t^{l-1}\mathbf{g}_t^K + \mathbf{M}_t[K-M]\mathbf{P}_t^M\mathbf{g}_t^K \tag{36}$$

Thus, its $\ell_2$-norm can be upper bounded via

$$\|\nabla\mathcal{L}_t(\boldsymbol{\theta}) - \hat{\nabla}^{\mathrm{Bi}}\mathcal{L}_t(\boldsymbol{\theta})\|$$
$$\leq \sum_{l=1}^{M}\left[\sum_{0\leq k_1<\ldots<k_L\leq K-1-l}\prod_{i=1}^{L}\alpha\|\mathbf{H}_t^{k_i}\|\right]\alpha\|\mathbf{H}_t^{K-l}\|\|\mathbf{P}_t^{l-1}\|\|\mathbf{g}_t^K\|+$$
$$\sum_{l=L+1}^{K-M}\sum_{0\leq k_1<\ldots<k_l\leq K-1-M}\left[\prod_{i=1}^{l}\alpha\|\mathbf{H}_t^{k_i}\|\right]\|\mathbf{P}_t^M\|\|\mathbf{g}_t^K\|$$
$$\leq \left[(\alpha H)^{L+1}\sum_{l=1}^{M}\binom{K-l}{L}(1-\alpha h)^{l-1} + \sum_{l=L+1}^{K-M}\binom{K-M}{l}(\alpha H)^l(1-\alpha h)^M\right]\|\mathbf{g}_t^K\|. \tag{37}$$

To this end, we next show that the two terms in (37) both decrease super-exponentially with $L$.

On one hand, plugging Lemma B.5 into the first term of (37) renders

$$(\alpha H)^{L+1}\sum_{l=1}^{M}\binom{K-l}{L}(1-\alpha h)^{l-1} < (\alpha H)^{L+1}\sum_{l=1}^{M}\left[\frac{e(K-l)}{L}\right]^L(1-\alpha h)^{l-1}$$
$$\overset{(a)}{\leq} (\alpha H)^{L+1}\left[\frac{e(K-1)}{L}\right]^L\sum_{l=1}^{M}(1-\alpha h)^{l-1}$$
$$= \frac{H}{h}\left[\alpha H\frac{e(K-1)}{L}\right]^L[1 - (1-\alpha h)^M]$$

where $(a)$ is because the index $l \geq 1$. This suggests there exists $C'_\alpha \leq e(K-1)/L$ to ensure its super-exponential diminish.

On the other hand, applying Lemmas B.7 and B.5 to the second term of (37) yields

$$
\sum_{l=L+1}^{K-M} \binom{K-M}{l}(\alpha H)^l(1-\alpha h)^M = (1-\alpha h)^M(\alpha H)^{L+1}\sum_{l=1}^{K-M-L}\binom{K-M-l}{L}(1+\alpha H)^{l-1}
$$

$$
\leq (1-\alpha h)^M(\alpha H)^{L+1}\sum_{l=1}^{K-M-L}\left[\frac{e(K-M-l)}{L}\right]^L(1+\alpha H)^{l-1}
$$

$$
\leq (1-\alpha h)^M(\alpha H)^{L+1}\left[\frac{e(K-M-1)}{L}\right]^L\sum_{l=1}^{K-M-L}(1+\alpha H)^{l-1}
$$

$$
= (1-\alpha h)^M\left[\alpha H\frac{e(K-M-1)}{L}\right]^L[(1+\alpha H)^{K-L-M}-1]
$$

which decreases super-exponentially with some $C'_\alpha \leq e(K-M-1)/L < e(K-1)/L$.  $\square$

**Lemma B.7.** *Let $\gamma \in \mathbb{R}$ be a constant. It holds for $L = 0, \ldots, K-1$ that*

$$
\sum_{l=L+1}^{K}\binom{K}{l}\gamma^l = \gamma^{L+1}\sum_{l=1}^{K-L}\binom{K-l}{L}(1+\gamma)^{l-1}. \tag{38}
$$

*Proof.* Defining the left-hand side of (38) as $S[K]$, it follows

$$
S[K] := \sum_{l=L+1}^{K}\binom{K}{l}\gamma^l \overset{(a)}{=} \sum_{l=L+1}^{K}\binom{K-1}{l-1}\gamma^l + \sum_{l=L+1}^{K-1}\binom{K-1}{l}\gamma^l
$$

$$
\overset{(b)}{=} \sum_{l=L}^{K-1}\binom{K-1}{l}\gamma^{l+1} + \sum_{l=L+1}^{K-1}\binom{K-1}{l}\gamma^l
$$

$$
\overset{(c)}{=} \binom{K-1}{L}\gamma^{L+1} + \sum_{l=L+1}^{K-1}\binom{K-1}{l}\gamma^{l+1} + \sum_{l=L+1}^{K-1}\binom{K-1}{l}\gamma^l
$$

$$
= \binom{K-1}{L}\gamma^{L+1} + \sum_{l=L+1}^{K-1}\binom{K-1}{l}\gamma^l(\gamma+1)
$$

$$
= \binom{K-1}{L}\gamma^{L+1} + (\gamma+1)S[K-1]
$$

$$
\overset{(d)}{=} \sum_{l=1}^{K-L+1}\binom{K-l}{L}\gamma^{L+1}(1+\gamma)^{l-1} + (1+\gamma)^{K-L-1}S[L+1]
$$

$$
= \gamma^{L+1}\sum_{l=1}^{K-L}\binom{K-l}{L}(1+\gamma)^{l-1}
$$

where $(a)$ is due to $\binom{K}{l} = \binom{K-1}{l-1} + \binom{K-1}{l}$ for $L+1 \leq l \leq K-1$ and $\binom{K}{K} = 1 = \binom{K-1}{K-1}$, $(b)$ changes the index in the first summation from $l$ to $l+1$, $(c)$ isolates the $l = L$ term from the summation, and $(d)$ leverages the relationship between $S[K]$ and $S[K-1]$ recursively.  $\square$

## C  NUMERICAL TEST SETUPS

All experiments are implemented on a desktop with an NVIDIA RTX A5000 GPU, and a server with NVIDIA A100 GPUs.

### C.1 SYNTHETIC DATA TEST

1-dimensional continuous sinusoid regression is performed, where the tasks are to fit the randomly sampled input values $x_t \in [-5, 5]$ to sinusoids $y_t = A_t \sin(x_t + \phi_t)$ of varying amplitudes $A_t \in [0.1, 5.0]$ and phases $\phi_t \in [0, \pi]$. During task-level training, the model is shown 10 data points. Since the purpose of this test is to view the meta-loss gradient accuracy of TruncMAML and BinomMAML, no meta-optimization was performed. For comparison, each method uses the same randomly generated dataset, ensuring parity among methods.

In Figure 3a, we set $K = 5$ and $L = 4$. In Figure 3b, the average errors are taken over 1,000 meta-batches, each containing 10 tasks.

### C.2 REAL DATA TEST

Real data tests are carried out on the miniImageNet (Vinyals et al., 2016) and tieredImageNet(Ren et al., 2018) datasets, which are both subsets of the ILSVRC-12 geared toward meta-learning. The *learn2learn* Python library (Arnold et al., 2020) was used to load the datasets. All images in both datasets are 3-channel RGB natural images cropped to $84 \times 84$ pixels. The datasets are divided as such:

- **miniImageNet:** 64 meta-train classes, 16 meta-validation classes, 20 meta-test classes. Each class contains 600 samples. This split was originally proposed in (Ravi & Larochelle, 2017).

- **tieredImageNet:** 351 meta-train classes (448,695 images), 97 meta-validation classes (124,261 images), and 160 meta-test (206,209 images). This split was originally proposed in (Ren et al., 2018).

For both datasets, the model architecture and training protocol is that suggested by (Vinyals et al., 2016; Finn et al., 2017), with 4 layers of $3 \times 3$ convolutions and 32 filters, followed by batch normalization, ReLU non-linearity, and $2 \times 2$ max-pooling. We sample our training and validation sets using the standard $W$-way $S^{\text{tr}}$-shot few-shot learning protocol. Specifically, for each mini-batch of classes, we randomly sample $W$ classes and draw $S^{tr}$ training images and $S^{val}$ validation images, totaling $W \times S^{\text{tr}}$ training and $W \times S^{\text{val}}$ training and validation images, respectively. We adopt the usual choices of $W = 5$ classes, $S^{\text{tr}} = 1$ or $S^{\text{tr}} = 5$ images, and $S^{\text{val}} = 15$ images. In addition, $K = 5$ adaptation steps are adopted. Meta-training are limited to $20,000$ iterations. Other hyperparameters follow from (Finn et al., 2017), i.e., adaptation step size $\alpha = 10^{-2}$, meta-training step size $\beta = 10^{-3}$, and meta batch size $B = 4$.

For numerical stability, we found it beneficial to replace the $\alpha$ in (8) with $\alpha' := L\alpha/K$. This substitution makes the condition $\alpha = \mathcal{O}((L+1)/(KH))$ in Theorems 3.8 and 3.10 more easily satisfied. It is worth stressing that $\alpha' = 0$ recovers FOMAML when $L = 0$, and $\alpha' = \alpha$ with $L = K$ reduces to MAML.

We also note that, current DL libraries such as PyTorch lack support for the niche but crucial ability to parallelize HVPs. Although the training gradients $\{\nabla \ell_t^{\text{trn}}(\phi_t^k)\}_{k=0}^{K-1}$ are obtained in the adaptation steps, our current implementation requires calculating them again in the backward pass of $\nabla \mathcal{L}_t(\boldsymbol{\theta})$. For a fair comparison, we have excluded the time for the this repeated gradient calculation.

## D EXTENSIONS AND FUTURE WORK

### D.1 HYBRID BINOM-TRUNC ESTIMATOR

Intuitively, in (4), the final training gradient $\mathbf{g}_t^K$ is less sensitive to second-order terms $\mathbf{H}_t^k$, with small $k$, from the earlier task-level training GD. In truncated backpropagation, this logic leads to the TruncGBML estimate (5), where $\mathbf{H}_t^k = \mathbf{0}_{d \times d}$, $0 \le k < K - L$. However, this intuition can also be applied to the BinomGBML estimate by similarly truncating the early Hessians and performing a binomial expansion on the remaining terms; thus, we arrive at a hybrid "binom-trunc" GBML estimate. To this end, the constant $C \in \{L, \ldots, K\}$ is introduced, which determines the truncation of the computational graph, with $L$ determining the polynomial order of the binomial expansion. The hybrid estimator can be constructed by setting $\mathbf{H}_t^k = \mathbf{0}_{d \times d}$, $0 \le k < K - C$, rendering

$$\hat{\nabla}^{\text{BiTr}} \mathcal{L}_t(\boldsymbol{\theta}) := \left[ \mathbf{I}_d + \sum_{l=1}^{L} \sum_{K-C \le k_{1:l} \uparrow < K} \prod_{i=1}^{l} (-\alpha \mathbf{H}_t^{k_i}) \right] \mathbf{g}_t^K. \tag{39}$$

Table 2: Hybrid binom-trunc estimate performance on miniImageNet and tieredImageNet with with $K = 5$, $L = 1$, and $C = 4$ and all other settings identical to that of tab. 1

|  | 5-way miniImageNet (%) | | 5-way tieredImageNet (%) | |
|---|---|---|---|---|
|  | 1-shot | 5-shot | 1-shot | 5-shot |
| Hybrid | $45.17 \pm 0.91$ | $62.41 \pm 0.49$ | $44.30 \pm 0.91$ | $63.78 \pm 0.49$ |
| Trunc | $44.53 \pm 0.90$ | $62.43 \pm 0.50$ | $44.67 \pm 0.96$ | $65.14 \pm 0.51$ |
| Binom | $45.50 \pm 0.91$ | $62.36 \pm 0.48$ | $46.23 \pm 0.95$ | $64.49 \pm 0.51$ |

The corresponding estimate can be constructed via alg. 1, setting $\nabla \ell_t^{\mathrm{trn}}(\phi_t^k) = \mathbf{0}_d$, $k = 0, \ldots, K-C$. Results on miniImageNet and tieredImageNet with $K = 5$, $L = 1$, and $C = 4$ are presented in table 2. Overall, the hybrid estimate performs slightly worse than TruncMAML and BinomMAML, which is expected since the constant $C$ introduces a trade-off between computational overhead and estimation accuracy.

## D.2 RELATION TO ANIL

As mentioned in section 3, akin to other meta-gradient estimates (TruncGBML, iMAML), our approach can be readily combined with ANIL (Raghu et al., 2019). Specifically, ANIL omits the task-specific adaptation of certain model layers, whereas BinomGBML can be used to efficiently estimate the meta-gradient of the remaining layers. While both ANIL and MAML have been shown effective on ImageNet-like datasets, such as those used for evaluation in section 4, it is up to the practitioner to choose the appropriate meta-learning algorithm.

