# OpenReview forum: "Binomial Gradient-Based Meta-Learning for Enhanced Meta-Gradient Estimation"
_ICLR.cc/2026/Conference — ICLR 2026 Poster_

### Official Review · Reviewer_4xo1 · 2025-10-26

**Soundness:** 4
**Presentation:** 3
**Contribution:** 3
**Rating:** 8
**Confidence:** 3

**Summary:**

This paper presents a meta-learning method BinomMAML, based on MAML, that uses the binomial expansion of a repeated (I - H_k) application in the meta-gradient to derive a parallelizable execution method and low-error approximation by truncating higher order terms in the expansion.  In contrast to truncating on iteration depth, all iterations' hessians can still be used in the approximation, with truncation happening on the degree of their combinatorial interactions.  This is efficiently computed by parallelizing the computation of the second-order values given the first, etc., so that computing the L'th order approximation requires L serialized sets of parallel computations.

**Strengths:**

* This is an insightful method whose basic derivation is explained well with clear motivation.
* Extensive theoretical derivations show its error decreases rapidly (possibly above exponential) and compare to similar derivations for MAML and Truncated MAML, showing excellent improvements.  These are verified on the (standard) sinusoid synthetic problem.
* The method also realizes gains are in the more real-world data settings of mini-Imagenet and tiered-Imagenet (albeit possibly less than one might hope for based on the theory), but still adequate empirical validation

**Weaknesses:**

* While the theoretical results are excellent, the method's results on the two real-data tasks is more limited and with relatively small gains. (though, the paper touches on this in the discussion and points out its gains are larger in the 1-shot case with reasonable explanation)

* Although I found the explanation of the method's ideas and the core derivation of the combinatorial algorithms and B operators clear, some of the formulas, and Alg. 1 in particular, are very busy and can be difficult to align with the derivation in Eq 8 and Prop 3.1.  Prop. 3.1 also has some indexing I found difficult to interpret in its use of $k_{L-i} = k_{L-1-i}+1$ as a sum subscript (though, I can see it corresponds to the combinatorial index variables 0<= k_1 < ...< k_l < K)

**Questions:**

Alg. 1 pseudocode I think could be a little better aligned to the notation in the text, in particular making use of the B operators and showing how the computations get these values.  The indexing is very busy in Alg. 1 and difficult to disentangle.

Figures and tables often appear far from their descriptions in the text and sometimes out of order.  It would be easier if this could be organized so the figures are closer to their main discussion.  For example, sec 4.1 on p8 uses Fig 3 on p.9, even though the next section 4.2 looks at the earlier Fig 2.  Fig 3 is also referenced for the first time earlier on p.4, which is OK, but I felt I had to jump around a little too much to find these several times.

---

> ### Author Response · Authors · 2025-11-21
>
> We thank the reviewer for their suggestions, which improved the quality and presentation of the work. We elaborate our responses to the reviewer's questions below.
>
> > Although I found the explanation of the method's ideas and the core derivation of the combinatorial algorithms and B operators clear, some of the formulas, and Alg. 1 in particular, are very busy....
>
> > **Q1** Alg. 1 pseudocode I think could be a little better aligned to the notation in the text, in particular making use of the B operators and showing how the computations get these values. The indexing is very busy in Alg. 1 and difficult to disentangle.
>
> Agreed, Alg. 1 can be difficult to connect with the text. Alg. 1 has been revised to a simplified version (previously in the appendix), and a diagram of the $B_t^{g_t^K,L-l}$ operator has been provided. These changes should help with clarity, particularly surrounding the indexing in Alg. 1.
>
> > **Q2** Figures and tables often appear far from their descriptions in the text and sometimes out of order. It would be easier if this could be organized so the figures are closer to their main discussion. For example, sec 4.1 on p8 uses Fig 3 on p.9, even though the next section 4.2 looks at the earlier Fig 2. Fig 3 is also referenced for the first time earlier on p.4, which is OK, but I felt I had to jump around a little too much to find these several times.
>
> Thanks for pointing this out. Figure placement has been altered to better match the flow of the text.

---

### Official Review · Reviewer_jYLW · 2025-10-30

**Soundness:** 3
**Presentation:** 2
**Contribution:** 2
**Rating:** 4
**Confidence:** 4

**Summary:**

In this paper, the authors aim to reduce the computational cost of meta-learning while maintaining a low approximation error. Existing approaches typically achieve efficiency gains at the expense of increased approximation bias. To address this trade-off, the authors propose a new framework called BinomGBML. They observe that conventional gradient-based meta-learning methods suffer from limited parallelism in the Hessian–vector product (HVP) computations. To overcome this limitation, the authors develop a meta-gradient estimator based on the binomial expansion, which expresses the meta-gradient as a truncated binomial series. This expansion is further reformulated into a cascade of vector operators, each of which can be executed in parallel. Theoretically, they show that the proposed estimator achieves a lower error bound compared to existing meta-learning methods. Empirically, they demonstrate the efficiency and effectiveness of BinomGBML through numerical experiments on multiple benchmark datasets.

**Strengths:**

1: The authors carefully investigate why existing variants of gradient-based meta-learning methods are inefficient. Their theoretical analysis provides valuable insights into the underlying causes of computational bottlenecks, helping the community better understand the limitations of prior approaches.

2: The proposed parallelized meta-gradient estimator is an interesting and original contribution. It introduces a new perspective on improving the efficiency of meta-learning by leveraging the binomial expansion to enable parallel computation of Hessian–vector products.

3: The method demonstrates a well-balanced trade-off between computational efficiency and gradient approximation accuracy, achieving significant speedups without sacrificing performance.

**Weaknesses:**

1: The method section is somewhat difficult to follow due to the dense mathematical formulation. Including high-level diagrams or flowcharts to illustrate the overall framework of BinomMAML would greatly enhance clarity and accessibility for readers.

2: As the proposed estimator introduces a new way to compute meta-gradients, the paper would benefit from a more detailed discussion on practical implementation — for instance, how to integrate the estimator into common frameworks like PyTorch and how to manage computational overhead in practice.

3:Although the paper emphasizes efficiency as a major advantage, it does not provide clear experimental results or ablation studies quantifying the computational gains. Presenting concrete benchmarks (e.g., runtime, memory usage, GPU utilization) would strengthen the empirical support for the proposed method.

**Questions:**

1: How can the proposed binomial-expansion based meta-gradient estimator be implemented in practice using PyTorch? A detailed explanation or example would help clarify how the estimator operates and how its components interact in a real training pipeline.

2: How does the runtime performance of the proposed estimator compare with existing baselines such as MAML, iMAML, and other meta-learning methods? Providing empirical comparisons or complexity analysis would help illustrate the computational advantages of the approach.

---

> ### Author Response · Authors · 2025-11-21
>
> We appreciate the reviewer’s constructive critique, and the opportunity it provided to revisit and refine our arguments. We address the points raised below.
>
> > **W1** The method section is somewhat difficult to follow due to the dense mathematical formulation. Including high-level diagrams or flowcharts to illustrate the overall framework of BinomMAML would greatly enhance clarity and accessibility for readers.
>
> Thanks for raising the concern. A diagram is now added in Fig. 1 to visualize the math formulation.
>
> > **Q1** How can the proposed binomial-expansion based meta-gradient estimator be implemented in practice using PyTorch? A detailed explanation or example would help clarify how the estimator operates and how its components interact in a real training pipeline.
>
> > **W2**  As the proposed estimator ... computational overhead in practice.
>
> Our code provides a good testbench for various meta-learning frameworks, including the proposed method, which can be found in \verb|binomial_maml.py| file in the supplementary materials. In short, the $K-L+1$ parallel HVPs per $B_t^{g,i}$ operator can be calculated by composing the parallelization function `torch.func.vmap()` with `torch.func.vjp()` and `torch.func.grad()`. This way, we need only store the intermediate parameters $\\{\\phi_t^{l}\\}_{l=0}^{K-1}$ to calculate the HVPs on the fly, thus bypassing the computational graph for the backward pass. The memory savings from this dynamic memory management are pointed out in Remark 3.4.
>
> > **Q2** How does the runtime performance of the proposed estimator compare with existing baselines such as MAML, iMAML, and other meta-learning methods? Providing empirical comparisons or complexity analysis would help illustrate the computational advantages of the approach.
>
> > **W3** Though the paper emphasizes efficiency as a ... empirical support for the proposed method.
>
> The paper presents empirical runtime and memory comparisons for TruncMAML, FOMAML, and MAML in figures 4(a,b). In the rebuttal, iMAML is added to these plots, as well as a new GPU utilization plot (fig. 4c). These plots corroborate the theoretical complexities states in sections 2 and 3, and demonstrate a strong case for BinomMAML in terms of computational efficiency.

---

### Official Review · Reviewer_UEik · 2025-10-31

**Soundness:** 3
**Presentation:** 4
**Contribution:** 4
**Rating:** 10
**Confidence:** 3

**Summary:**

This paper studies how to obtain accurate estimates of the multi-step inner-optimization gradient in gradient-based meta-learning in both fast and memory efficient manner. In MAML, a representative approach to meta-learning, estimating the posterior over task-invariant parameters requires multiple steps of gradient computation, which becomes a major computational bottleneck. To mitigate this, prior work has used truncation of the number of steps that contribute to the gradient approximation or implicit differentiation, yet these approaches suffer from gradient estimation error and numerical instability. This paper shows that, if one replaces a naive decomposition of the multi-step gradient with a decomposition inspired by the binomial expansion when truncating the number of steps, the truncation error decays exponentially with respect to the inner step size. The proposed method can be algebraically expressed as an operator acting on the gradient vector, which yields a parallelizable algorithm. Across several settings including sufficiently realistic ones, the authors demonstrate both theoretically and empirically that the proposed method attains smaller gradient estimation error than existing methods. In addition, as a secondary contribution, they show in theory and experiments that the proposed method reduces space complexity even relative to vanilla MAML.

**Strengths:**

- The problem is very clearly formulated and is important to the community. Gradient-based meta-learning exemplified by MAML is widely studied, and obtaining accurate gradients while controlling computation is obviously one of the central concerns.
- The paper is well written and easy to follow. In particular, it organizes gradient-based meta-learning from a consistent perspective. The target problem is framed via hierarchical Bayesian modeling with MAML positioned as one solution, and the paper provides a comprehensive review of representative methods for addressing computation time, including FO-MAML, Reptile, Truncated MAML, and implicit MAML. This gives readers a coherent viewpoint and effectively motivates the proposed method.
- The idea inspired by the binomial expansion solves the problem elegantly with a solid construction. While it does not aim to remove the fundamental growth in computation with the number of steps as implicit MAML does, it is highly practical. The result is a gradient truncation scheme whose error decreases exponentially with the inner step size and that admits parallel computation, which is a noteworthy contribution.
- The paper provides consistent proofs of upper bounds on gradient estimation error under multiple scenarios, from sufficiently realistic assumptions up to the stronger assumption of convex losses. These theoretical results strongly support the intuition of the method.
- The theoretical results appear sound, although I could not verify every proofs in full detail.
- The authors provide numerical validation under realistic settings for gradient estimation accuracy, time complexity, and space complexity. The experiments show that BinomMAML consistently outperforms other methods in gradient accuracy, while the time overhead remains acceptable even compared with TruncMAML. Moreover, although secondary, improved memory efficiency compared with vanilla MAML is also a significant advantage.

**Weaknesses:**

- Although implicit MAML is discussed as prior work, the paper did not provide any of theoretical or empirical comparisons to it. Since iMAML is undoubtedly one of the representative approaches to addressing MAML’s computational burden, would it be possible to include performance comparisons with such methods, either theoretically or experimentally, to validate the effectiveness of the proposal?
- Theoretical claims from Proposition 3.1 through Theorem 3.10 are central and valuable contributions. However, there is concern that the proofs of these statements are not discussed in the main text at all. For core results such as Proposition 3.1, Theorem 3.2, and Theorem 3.6, even a high-level proof sketch in the main body would substantially aid reader understanding.

**Questions:**

- The proposal appears to conceptually distinguish BinomGBML from BinomMAML, but the practical difference is not clear to me. Chapter 3 seems to assume the MAML problem setting described in Chapter 2. Is this understanding correct?
- What is the Hybrid Binom-Trunc Estimator section in Appendix D for?

---

> ### Author Response · Authors · 2025-11-21
>
> We sincerely thank the reviewer for engaging deeply with the manuscript and contributing thoughtful recommendations. We address specific points raised by the reviewer below.
>
> > **W1** Although implicit MAML is discussed as prior work, the paper did not provide any of theoretical or empirical comparisons to it. Since iMAML is undoubtedly one of the representative approaches to addressing MAML’s computational burden, would it be possible to include performance comparisons with such methods, either theoretically or experimentally, to validate the effectiveness of the proposal?
>
> iMAML's performance on miniImageNet and tieredImageNet are included in Table 1. For better context surrounding iMAML, we add an explanation of iMAML's space and time complexity in the preliminaries, as well as comparing its empirical space and time complexity in the results (fig. 4 in the revision).
>
> > **W2** Theoretical claims from Proposition 3.1 through Theorem 3.10 are central and valuable contributions. However, there is concern that the proofs of these statements are not discussed in the main text at all. For core results such as Proposition 3.1, Theorem 3.2, and Theorem 3.6, even a high-level proof sketch in the main body would substantially aid reader understanding.
>
> Agreed, it may help the reader to have a high-level idea of the proofs in the body. In the revision, we provide the following sketch of the core proposition 3.1:
> > The proof is carried out by induction on $L$. The base case $L=1$ is readily shown by applying the definition of $B_t^{i}$. Assuming the cases $L=1,\dots,L'$ (where $L'<K$) hold, the sums of (9) can be recursively expanded and rewritten in terms of $B_t^{i}$ to obtain
>
> $$
> I_d + \\sum_{l=1}^{L'+1} \\sum_{0\\le k_{1:l} \\uparrow < K} \\prod_{i=1}^l (-\\alpha H_t^{k_i}) = B_t^{L'} B_t^{L'-1} \\ldots B_t^{1} \Bigg[ I_d + \\sum_{k_{L'+1}=k_{L'}+1}^{K-1} (-\\alpha H_t^{k_{L'+1}}) \\Bigg]
> $$
>
> $$
>     ~~~~~~~~~~~~~~~~~~~~~~~~~~~~~~~~~~~~~~~~~~~~~~~~~~ =  B_t^{L'} B_t^{L'-1} \\ldots B_t^{1} B_t^{0} I_d
> $$
>
>
> > **Q1** The proposal appears to conceptually distinguish BinomGBML from BinomMAML, but the practical difference is not clear to me. Chapter 3 seems to assume the MAML problem setting described in Chapter 2. Is this understanding correct?
>
> Yes, lines 207-209 point out the same setting is assumed in Sections 2 and 3. BinomMAML, which is considered a special instance of BinomGBML, is developed as a running paradigm for clarity and simplicity, rather than considering every GBML algorithm.
>
> > **Q2** What is the Hybrid Binom-Trunc Estimator section in Appendix D for?
>
> We thank the reviewer for pointing this out. This section was omitted by error but is included in the revised draft. It proposes a hybrid estimate based on BinomGBML, where the computational graph is further truncated to the final $C$ GD steps as in TruncGBML, for some $C\in\{L,\dots,K\}$. Further details are provided in the revised draft.

---

> > ### Comment · Reviewer_UEik · 2025-11-27
> >
> > Thank you very much for the further clarification. I remain convinced that this paper is significant, so I will keep my initial score.

---

### Official Review · Reviewer_r31s · 2025-11-04

**Soundness:** 3
**Presentation:** 4
**Contribution:** 3
**Rating:** 6
**Confidence:** 4

**Summary:**

Gradient based meta-learning (GBML) suffers from high computational overhead. While approximations have been proposed to make GBML more tractable, these approximations suffer from large errors. This work proposes BinomMAML via targeted approximations achieved via truncated binomial expansion yields better meta-gradient estimation, in a efficient and scalable manner. The authors present theoretical analysis of their approximation and empirically validate its efficacy in comparisons with a range of competitive baselines.

**Strengths:**

1) The truncated binomial expansion along with the  regrouping terms into a few parallelizable computations is quite appealing
2) The regrouping of terms enables parallel computation, increasing GPU usage and overall efficiency.
3) They show quantitive improvements over standard approximation methods

**Weaknesses:**

1) Concerns about benchmark choice. Prior work (e.g., ANIL/NIL [1]) shows that on ImageNet-style few-shot benchmarks, removing or nearly removing the inner loop can match MAML’s performance—likely because train and test tasks are sampled from the same ImageNet distribution, so distribution shift is limited. If inner-loop adaptation offers little measurable benefit in this regime, then demonstrating a “better” meta-gradient approximation there is a weak proxy for approximation quality: the task may not actually require accurate meta-gradients. Consequently, results on ImageNet/TieredImageNet alone are hard to interpret in terms of the value of improved meta-gradient estimation.

2) Lacks discussion of [1].

[1]https://arxiv.org/abs/1909.09157

**Questions:**

1) I would suggest comparing the current approach to ANIL or NIL on cross domain meta learning settings or other challenging meta learning settings where there is a significant distribution shift between training and testing task distributions [1], table 3 in [2].


[1] https://openaccess.thecvf.com/content/CVPR2023W/VISION/papers/Lee_XDNet_A_Few-Shot_Meta-Learning_Approach_for_Cross-Domain_Visual_Inspection_CVPRW_2023_paper.pdf

[2] https://arxiv.org/pdf/1904.04232

---

> ### Author Response · Authors · 2025-11-21
>
> We thank the reviewer for their valued perspective and insight. Below, we address the points raised in the review.
>
> > **W1** Concerns about benchmark choice. Prior work (e.g., ANIL/NIL [1]) ... improved meta-gradient estimation.
>
> > **Q1** I would suggest comparing the current approach to ANIL or NIL on cross domain meta learning settings or other challenging meta learning settings where there is a significant distribution shift between training and testing task distributions.
>
> Akin to other meta-gradient estimates (TruncGBML, iMAML), our approach can be readily combined with ANIL; the methods are complementary. Specifically, ANIL omits the task-specific adaptation of certain model layers, whereas BinomGBML can be used to efficiently estimate the meta-gradient of the remaining layers. While both ANIL and MAML have been shown effective on ImageNet-like datasets, it is up to the practitioner to choose the appropriate meta-learning algorithm for their given dataset.
>
> > **W2** Lacks discussion of [1].
>
> The following has been added to Appendix D:
> > As mentioned in section 3, akin to other meta-gradient estimates (TruncGBML, iMAML), our approach can be readily combined with ANIL [1]. Specifically, ANIL omits the task-specific adaptation of certain model layers, whereas BinomGBML can be used to efficiently estimate the meta-gradient of the remaining layers. While both ANIL and MAML have been shown effective on ImageNet-like datasets, such as those used for evaluation in section 4, it is up to the practitioner to choose the appropriate meta-learning algorithm.
>
> [1]https://arxiv.org/abs/1909.09157

---

### Author Response · Authors · 2025-11-21
**Summary of Revision**

First, we sincerely thank the reviewers for their thoughtful questions and important points raised about the paper. The following changes have been made in the revision:

- One of the main concerns seems to be the dense mathematical formulation, and in particular, the interpretability of Alg. 1. To this end, we rewrite Alg. 1 using a formulation previously found in Appendix A that more directly relates to the core operator $B_t^{g,i}$, and we include a new figure (fig. 1 in the revision) illustrating the $(L-l+1)$-th step operator $B_t^{g_t^{K},L-l}$.
- Another concern raised was about a more complete comparison to iMAML. To address this, we briefly elaborate the theoretical time and space complexity of iMAML in lines 187-189 in the revision, and we expand Fig. 4 to include empirical complexity results of iMAML.
- To more completely compare the compute cost of each algorithm, a new plot detailing the GPU compute core utilization of each algorithm is added to Fig. 4.
- The caption on Table 1 has been updated to say ``standard deviation'' rather than ``95\% confidence interval'' as was erroneously stated in the original draft.
- A proof sketch of the core Proposition 3.1 has been added.
- Appendix D has been added, including a short subsection detailing a hybrid Binom-Trunc estimate, and another subsection detailing the relation of BinomGBML to ANIL.
- Minor wording fixes.
- Lastly, figures have been enlarged for clarity.

Again, we are grateful for the insight of the reviewers and welcome conversation during the discussion period.

---

### Meta-Review · Area_Chair_4j4n · 2026-01-09

**Summary:**

The main concerns were really only those of jYLW, on dense math formulation and a few more experiments to provide.

**Reviewer Concerns:**

The concerns on the additional experiments of jYLW were fulfilled.

**Reviewer Scores:**

All other reviews having positive polarity, it is far to assume jYLW would have eventually breaking the somehow borderline tie by moving on positive opinion.

---

### Decision · Program_Chairs · 2026-01-26

Accept (Poster)